

# Development of a total variation diminishing (TVD) Sea ice transport scheme and its application in in an ocean (SCHISM v5.11) and sea ice (Icepack v1.3.4) coupled model on unstructured grids

Qian Wang[1,2], Fei Chai[1,2,3], Yang Zhang[2], Y. Joseph Zhang[4], Lorenzo Zamperi[5]

1. School of Oceanography, Shanghai Jiao Tong University, Shanghai, China

2. State Key Laboratory of Satellite Ocean Environment Dynamics, Second Institute of Oceanography, Ministry of Natural Resources, Hangzhou, China

3. State Key Laboratory of Marine Environmental Science, Xiamen University, Xiamen, China

4. Virginia Institute of Marine Science, Gloucester Point, VA, USA

5. Climate Simulations and Predictions Division, Centro Euro-Mediterraneo sui Cambiamenti Climatici (CMCC), Bologna, Italy

*Correspondence to*: Fei Chai (fchai@sio.org.cn ) Yang Zhang (yzhang@sio.org.cn )

**Abstract.** As the demand for increased resolution and complexity in unstructured sea ice models is
growing, a more advanced sea ice transport scheme is needed. In this study, we couple the Semi-implicit Cross-scale Hydro-science Integrated System Model (SCHISM) with Icepack, the column physics package of the sea ice model CICE; a key step is to implement a total variation diminishing (TVD) transport scheme for the multi-class sea ice module in the coupled model. Compared with the upwind scheme and a central difference scheme, the TVD transport scheme is found to have better performance
for both idealized and realistic cases, and meets the requirements for conservation, accuracy, efficiency (even with very high resolution), and strict monotonicity. The coupled model for the Arctic Ocean successfully reproduces the long-term changes in the sea ice extent, the sea ice boundary and concentration observation from the satellite.

## 1 Introduction

The dramatically decrease in the Arctic Sea ice in recent decades due to global warming has a major

impact on local and global climate (IPCC, 2019). In order to understand the changes in the physical and

biogeochemical processes occurring in the Arctic Ocean, numerical models have become an important

tool and they have been significantly improved in the past few decades. The sea ice, as a highly

complex material (Hunke et al., 2020), received special attention, and the sea ice models have become

more sophisticated in representing realistic physics. At present, an advanced sea ice model, the Los

Alamos sea ice model (CICE, Hunke et al., 2015), including a stand-alone column physics package

Icepack (Hunke et al., 2020), has incorporated multi-class thermodynamics, such as the Bitz and





Lipscomb (1999; BL99) thermodynamics formulation for constant salinity profiles, the mushy layer

thermodynamics formulation for evolving salinity (Turner et al. 2013), and the sea ice ridging

processes (Hunke, 2010). Many structured-grid models have been coupled with Icepack or CICE

directly or via couplers, e.g., the Community Earth System Model (CESM, Hurrell et al.,2013) and the

HYbrid Coordinate Ocean Model (HYCOM); others have partially incorporated and adapted CICE

subroutines in their own ice module, e.g., the Sea Ice modelling Integrated Initiative ($SI^3$) of the

Nucleus for European Modelling of the Ocean (NEMO, and NEMO can also couple with CICE or The

Louvain-La-Neuve sea ice model, LIM3, Gurvan et al., 2022), and The Thermodynamic Sea Ice

Package (THSICE) of the Massachusetts Institute of Technology General Circulation Model (MITgcm,

Campin et al. 2023). For the unstructured-grid (UG) models, Gao et al. (2011) have incorporated the

Unstructured-Grid CICE (UG-CICE) into the unstructured-grid Finite Volume Community Ocean

Model (FVCOM, Chen et al., 2012). Some other unstructured-grid models have incorporated Icepack

directly, e.g., the Finite-volumE Sea ice-Ocean Model version 2 (FESOM2, Danilov et al., 2017) and

the Model for Prediction Across Scales (MPAS-Seaice, Turner et al., 2022). Note that when Icepack is

incorporated into another model, the latter must also implement its own dynamic solver for momentum

and transport. UG-CICE and FESOM2 use triangular mesh grids while MPAS-Seaice uses Voronoi

dual graph. UG-CICE uses a finite-volume formulation, the sea ice component allows for five ice

categories, four layers of ice and one layer of snow. The other models allow users to specify the

number of categories. UG-CICE can produce good results on the seasonal variability of the sea ice in

the Arctic Ocean (Gao et al., 2011). FESOM2 has implemented Icepack in its entirety and found that

more complex model formulations lead to better results (Zampieri et al., 2021). MPAS-Seaice can be

viewed as the unstructured version of CICE, and thus shares sophisticated thermodynamics and

biogeochemistry with CICE, including BL99 and mushy layer, and is the current sea-ice component of

the Energy Exascale Earth System Model (E3SM, Turner et al., 2022). A summary of these sea-ice

models is given in Table 1.

| Model | Ice model | Grid | Thermodynamic | Transport solver | Coupling method |
|---|---|---|---|---|---|
| CESM | CICE | Structured | BL99<br>Mushy Layer | Incremental remapping scheme/<br>Upwind scheme | Coupler |
| NEMO | CICE, LIM3, $SI^3$ | Structured | Mushy Layer | Prather scheme/<br>ULTIMATE-MACHO scheme ($SI^3$) | Direct (with $SI^3$) |
| HYCOM | CICE | Structured | BL99<br>Mushy Layer | Incremental remapping scheme/<br>Upwind scheme | Coupler |



| | | | | | |
|---|---|---|---|---|---|
| MITgcm | THSICE | Structured | Two layers of ice and one layer of snow | 2nd-order flux limited scheme | Direct |
| E3SM | MPAS-Seaice | Unstructured | BL99 Mushy Layer | Incremental remapping scheme/ Upwind scheme | Coupler |
| UG-CICE | CICE | Unstructured | Four layers of ice and one layer of snow | Second order upwind scheme | Direct |
| FESOM2 | ICEPACK | Unstructured | BL99 Mushy Layer | FEM-FCT | Direct |

**Table 1.** Comparison of several sea ice models

With the advancement in High Performance Computing, sea ice coupled models are increasingly
executed on higher spatial and temporal resolutions. The increased demand for resolution and complexity
in the sea ice models calls for an accurate, stable, conservative, strictly monotonic, and efficient sea ice
transport method (Hunke et al. 2010). The sea ice transport method has been studied for many years, and
various schemes have been proposed. Lipscomb et al. (2004) implemented the upwind and incremental
remapping schemes in CICE, both of which are still available in the latest version. The upwind scheme
is the simplest scheme for transport, but it is too diffusive due to its first order accuracy. The incremental
remapping scheme is a second-order accurate scheme, and has great performance in structured grid
models, but is inefficient for highly distorted UGs. For example, MPAS-Seaice uses the incremental
remapping scheme (Turner et al., 2022), but as a global model, its resolution is usually coarse. MITgcm
offers many tracer advection solvers, but it recommends flux-limited schemes to avoid unphysical results
(Campin et al. 2023). NEMO uses the Prather scheme or the ULTIMATE-MACHO scheme with $SI^3$ ice
model (Gurvan et al., 2022), both of which require some functions to limit the tracer concentrations from
exceeding the largest values of all adjacent nodes. The efficiency has not been well tested on those SG
models under very high resolution down to about tens of meters.

In the case of triangular UGs, the transport scheme utilized in UG-CICE is the second-order upstream
scheme, which considers the gradient of sea ice tracers (Gao et al., 2011). This scheme is consistent with
the tracer transport in FVCOM (Chen et al., 2012). It is unclear if the monotonicity is guaranteed by this
scheme or if additional diffusion is needed. The transport scheme of FESOM2 is the Finite Element Flux
Corrected Transport scheme (FEM-FCT, Löhner et al., 1987), which is based on the finite element
description (Danilov et al. 2015). It is also a conservative and second-order scheme (Budgell, et al. 2007),
but its cost is linearly increasing with the number of variables, and more importantly, strict monotonicity
comes with a higher cost (Löhner et al., 1987). Therefore, Zhang et al. (2023) used the upwind scheme
by zeroing out the higher-order contribution in their study with very high resolution.

SCHISM, which has the seamless cross-scale capability from creek to ocean (Zhang et al., 2016), has
been applied to study the Great Lake ice formation process and obtained reasonable results in very high



resolution (Zhang et al., 2023), using a single-class ice/snow module borrowed from FESOM (Danilov

et al., 2015). The thermodynamics employed is a 0-layer thermodynamic module (Parkinson &

Washington, 1979), with constant dry and melting albedos of ice and snow. In the simulation of The

Great Lake ice formation process, both SCHISM and single-class ice model allow multi-scale physics

on variable resolution (Zhang et al., 2023). However, the performance of the multi-class sea ice

formulation has not been tested before. SCHISM therefore represents a mature and reliable platform to

implement the multi-class sea ice module, Icepack.

This paper presents a new unstructured ice-ocean coupled model built on SCHISM and Icepack. The

coupled model utilizes the TVD transport scheme to achieve an efficient, strictly monotone, second-order

accuracy scheme for ice tracers on generic unstructured grids (even with locally very high resolution).

Section 2 introduces components of the coupled model and describes how the TVD transport scheme is

implemented for the ice model. In Section 3, we compare some ideal test results from the new TVD

scheme with two other methods (the upwind scheme and a central difference scheme) and compare the

efficiency with upwind scheme when applied to a high-resolution mesh; we also validate the new coupled

model with a simulation of the Arctic Ocean sea ice with realistic atmosphere forcing. Section 4 compares

the new TVD scheme with other TVD schemes. Section 5 summarizes the major findings of this work.

## 2 Method

### 2.1 Icepack implemented in SCHISM

We couple Icepack v1.3.4 (Hunke et al., 2023) with SCHISM v5.11. Besides the 0-layer

thermodynamics, two more sophisticated thermodynamic formulations, BL99 and the mushy layer are

also implemented. At the sub-grid scale, thin and thick ice coexist, and therefore an ice thickness

distribution (ITD, Bitz et al.,2001) has been implemented in order to describe the unresolved spatial

heterogeneity of the thickness field. The ITD provides a prognostic statistical description of the sea ice

thickness partitioned into multiple categories and of the ice area fraction associated to each category,

instead of only one fraction as in the previous implementation. More traces and more ice processes are

added in this new version of Icepack, including multiple melt ponds parameterizations (Hunke et al.,

2013) and a mechanical redistribution parameterization (Hunke 2010) that responds to sea ice

convergence by piling up thin sea ice and therefore mimicking ridging and rafting events. The interaction



between the shortwave radiation and the sea ice in Icepack is described by the 'Community Climate System Model (CCSM3)' formulation, which links the surface albedo to the surface sea ice temperature, or the Delta-Eddington formulation (Briegleb et al., 2007), which links the albedo to inherent optical properties of sea ice and snow. The dynamic solver is not included in Icepack and is based on two approaches: 1. a classic Elastic-Viscous-Plastic method (EVP, Hunke & Dukowicz, 1997), and 2. modified Elastic-Viscous-Plastic method (mEVP, Kimmritz et al., 2015), both inherited from the old single-class ice/snow formulation (Zhang et al. 2023). All ice-related subroutines are called at every ocean step by SCHISM's hydrodynamic core. The ice module exports to SCHISM variables needed for coupling such as the shortwave radiation, the ice-ocean heat flux, the freshwater flux, and finally the sea ice pressure and ice-ocean stress for all ice-covered nodes, in proportion to the sea ice area fraction. Over open ocean these variables are calculated directly by SCHISM. All variables required by Icepack can be obtained from either SCHISM or separate input files.

## 2.2 Schemes for sea ice transport

The basic transport equation of sea ice area or fraction $a_n$ for each sea ice category is (Thorndike et al. 1975),

$$\frac{\partial a_n}{\partial t} + \frac{\partial u a_n}{\partial x} + \frac{\partial v a_n}{\partial y} + \frac{\partial}{\partial h}(a_n f) = \psi, \tag{1}$$

Where $u$ and $v$ are the ice velocities of $x$ and $y$ components, respectively, and $h$ is the ice thickness. The last term on the left side is thermodynamic change, where $f$ is the rate of ice melting or growing, and the right-side term $\psi$ is mechanical redistribution like the ridging process. We solve this equation using a fractional step method: first solve a pure advection equation (i.e. by setting the thermodynamic term and mechanical redistribution term to 0), followed by a correction step that includes the remaining terms. The main challenge occurs in the first step, where we must solve a pure advection equation for one category of sea ice fraction $a_n$:

$$\frac{\partial a_n}{\partial t} + \frac{\partial u a_n}{\partial x} + \frac{\partial v a_n}{\partial y} = 0, \tag{2}$$



Note that the ice velocity field is divergent, which can produce new local maxima/minima. However, a

strictly monotone scheme is still desirable in order to separate the numerical dispersion from the physical

convergence.

We apply a finite volume algorithm to discretize equation (2). Unlike the Arakawa-CD grid used in

SCHISM, the sea ice module inside SCHISM employs an Arakawa-A grid, with both the sea ice velocity

and traces located at the node (blue circles in Fig.1). The tracer control volume is defined as the polygon

enclosed by the lines composed of centroids and edge centers (red circles in Fig.1). So, in the subsequent

time step, after $\Delta t$, the new ice fraction is:

$$a_n^{t+1} = a_n^t + \frac{\Delta t \sum_{i \in S} Q_i \phi_i}{\Omega_S}, \tag{3}$$

$\Omega_S$ is the total area of the control volume; $S$ is its boundary, and $Q_i$ is the flux across the edge $i$ of the

control volume. Most of these variables can be obtained easily in the model, so we only focus on finding

a method to proximate the edge tracer value, $\phi_i$.

The simplest method is the first order upwind scheme (assuming, without loss of generality, the velocity

direction is as depicted in Fig. 1):

$$\phi_i = \phi_C, \tag{4}$$

The central difference scheme is

$$\phi_i = \frac{1}{2}(\phi_C + \phi_D), \tag{5}$$

Where $\phi_C$ and $\phi_D$ are the values at the upwind and the downwind nodes, respectively, for one edge of

the control volume (Fig.1a).

The TVD corrects the upwind values as:

$$\phi_i = \phi_C + \frac{\psi_i}{2}(\phi_D - \phi_C), \tag{6}$$

The last term on right side is the anti-diffusion correction. In this part, $\psi_i$ is a function of the upwind

ratio, $r_i$, for which we select the Van-leer limiter (van Leer, 1979),

$$\psi_i = \frac{r_{i*} + |r_i|}{1 + |r_i|}, \tag{7}$$

$$r_i = \frac{\phi_C - \phi_{U*}}{\phi_D - \phi_C}, \tag{8}$$

Where $\phi_{U*}$ is defined as the upwind node of the upwind node (i.e., 'up-upwind'), and can be accessed

easily in a structured grid or a uniform unstructured grid (Fig.1a). But for generic unstructured grids, how

to approximate $\phi_{U*}$ is a key issue for the TVD scheme. There are several possible choices for $\phi_{U*}$, and





after some comparisons we choose the method proposed by Darwish et al. (2003). This method includes

the gradient of the central node $\nabla\phi_C$.

$\phi_{U*} = \phi_D + R_{DU} \cdot (\nabla\phi_C) = \phi_D - 2R_{CD} \cdot (\nabla\phi_C),$    (9)

where $R_{DU}$ is the vector from the downwind node to the up-upwind node, and $R_{CD}$ is the vector from

the upwind to downwind nodes.

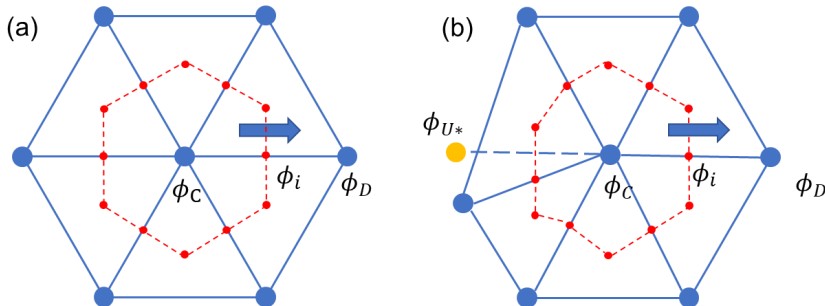

**Figure 1.** Schematics of control volume for the ice transport; **(a)** is for a uniform unstructured mesh, **(b)** is for a generic
unstructured mesh.

As the sea ice concentration cannot exceed 1 or be negative in this pure advection step (but after the

transport step, it can exceed 1 and lead to the ridging process, and in the latter case, Icepack will perform

clipping), Darwish's method (9) can produce errors and needs to be limited:

$\phi_{U*} = \min\big(1, max\big(0, \phi_D - 2R_{CD} \cdot (\nabla\phi_C)\big)\big),$    (10)

Using the approximation of edge tracer values $\phi_{U*}$, we can calculate the sea ice fluxes across every

edge of the control volume, and thus the new concentration from Eq. (3). Other tracer fluxes like

volume per unit area of ice and enthalpy depend on the area fluxes, as does CICE. For instance, the

volume per unit area of ice $v_n$ equals the product of the sea ice area $a_n$ and the sea ice thickness $h_n$,

here $h_i$ is the sea ice thickness of the upwind node.

$v_n = a_n h_n,$    (11)

$v_n^{t+1} = v_n^t + \frac{\Delta t \sum_{i \in S} Q_i \phi_i h_i}{\Omega_S},$    (12)

Sea ice enthalpy $e_n$ is the product of the sea ice area $a_n$, the sea ice thickness $h_n$ and the energy per

unit volume $q_n$, here $q_i$ is the energy per unit volume of upwind node.

$e_n = a_n h_n q_n,$    (13)

$e_n^{t+1} = e_n^t + \frac{\Delta t \sum_{i \in S} Q_i \phi_i h_i q_i}{\Omega_S},$    (14)



Other tracers at the new step can be calculated this way.

The finite-volume method ensures global and local conservation of tracers. Given that all ice area fluxes are recorded, flux calculation within the same category of ice only needs to be done once, so the method

is computationally efficient. Numerous tests have demonstrated that the TVD scheme provides second-order accuracy in smooth regions (Zhang et al., 2015), and guarantees strict monotonicity and a good accuracy. The limiter we chose is the widely used Van-leer limiter. Even though the accuracy of this limiter may be locally reduced to first order, it always maintains monotonicity as long as the time step used satisfies the stability condition, as demonstrated by Sweby (1984). Sea ice concentration may

produce new extremes after the transport step due to convergence. However, it should never be negative; this is guaranteed because the method in Eq. (3) is essentially a weighted average method with non-negative weights.

## 3 Result

### 3.1 Idealized test case

Since the thermodynamic part and dynamic parts of this model are relatively mature and have been widely utilized in other models, in this study we focus on validating the new transport scheme. The comparison of a few transport schemes is carried out through an idealized ice transport experiment in a uniform unstructured mesh. The mesh grid consists entirely of equilateral triangles, with a side length of 200m for each triangle. As the initial condition, we placed a rectangular sheet of sea ice with dimensions

of 5000m x 5000m on the left side of the mesh. The initial ice thickness is 1.5m, and it moves to the right along the x-axis at a speed of 1m/s. The time step is 1 second, which satisfies the Courant-Friedrichs-Lewy condition of TVD (Zhang et al., 2016). We run the idealized experiment for 24 hours, or 86400 steps. We select two other schemes for comparison, the first order upwind scheme and the central difference scheme (with proper limiting based on local max/min). The skill metrics include the accuracy,

conservation, and monotonicity of the results.

### 3.1.1 Accuracy

Fig. 2 shows the snapshots and central profiles along the x-axis of the sea ice concentration taken every 3 hours, with the theoretical solution represented by the red rectangles. For clarity, only areas with a concentration greater than 15% are shown. Compared to other schemes, the upwind scheme is



significantly more diffusive, yet relatively uniform. The outline varies in shape, transitioning from a

square to a circle. At the end of the model run, the peak of ice concentration is approximately 30% of the

initial value, which is unsurprisingly the lowest among the three schemes. The central difference scheme

retains more sea ice in the red rectangle than the upwind scheme. However, it produces non-uniform

results even though the ice speed is uniform, and it requires clipping of under/overshoots (which violates

the conservation). Some sea ice is left behind, with a banded distribution along the *x*-axis (Fig. 2b). The

profiles portrayed in Fig. 2e indicate that the sea ice concentration exhibits multiple peaks while the peak

ice concentration reaches approximately 90% in the end. Figs. 2c and 2f demonstrate that the TVD

scheme has the best accuracy compared to the other two schemes. The horizontal distribution of the ice

is closest to the analytical solution and exhibits a peak ice concentration around 100% at all times, despite

some minor diffusion on the frontal edges.

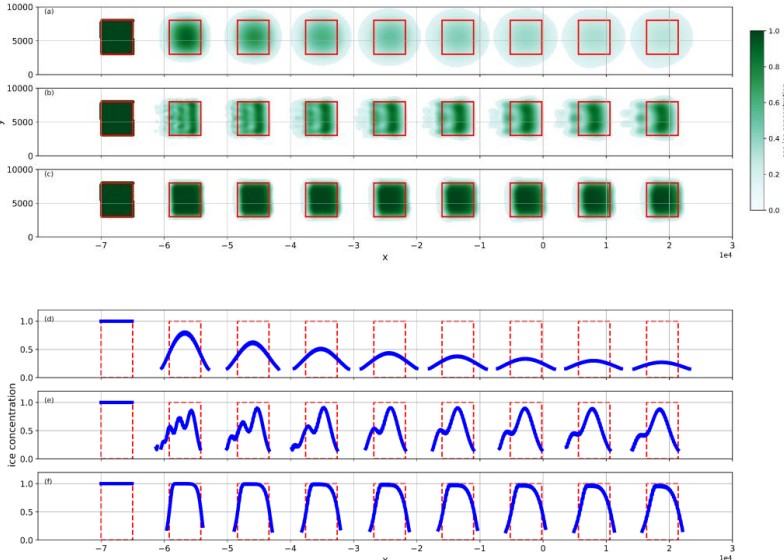

**Figure 2.** Sea ice concentration snapshots **(a-c)** and profiles **(d-f)**. The sea ice moves from left to right, snapshots
are taken every 3 hours, and the red rectangular is the exact solution. **(a, d)** Upwind, **(b, e)** Central difference, **(c, f)**
TVD.

The results of the ice volume per unit area are similar to the ice concentration (Fig. 3). The upwind

scheme is the most diffusive one. The peak of ice volume per unit area is only 0.4 meters at the end. The

central difference scheme is superior to the upwind scheme, and there is no excessive amount of ice that

lies outside the red rectangle on the top, bottom, and front edges. However, the scheme still shows



multiple peaks in Fig. 3e. Furthermore, it has obvious overshooting in all snapshots, which can be attributed to some spurious convergence processes that should not have occurred. The TVD scheme is the best of the three schemes. It performs well in terms of both shape and peak value. The shape of the ice volume per unit area is close to the theoretical solution from start to finish, with the peak value slightly

(1.496m) lower than the exact result at the end. The ice volume from the upwind and TVD (Fig. 3d and f) has similar shapes to the ice concentration (Fig. 2d and f), indicating that both schemes maintain the monotonicity well, and we will discuss more on this later.

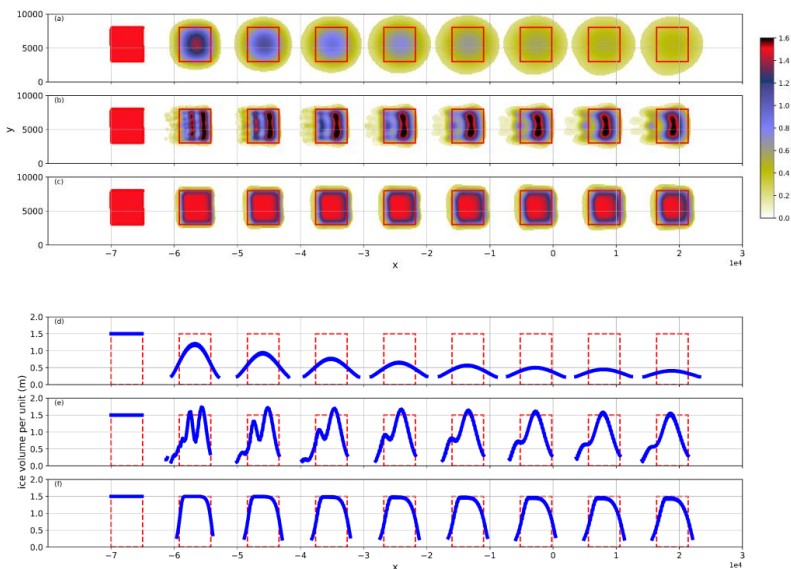


**Figure 3.** Volume per unit area of ice, with snapshots **(a-c)** and profiles **(d-f).** The sea ice moves from left to right, snapshots are taken every 3 hours, and the red rectangular is the exact solution. **(a, d)** Upwind, **(b, e)** Central difference, **(c, f)** TVD

### 3.1.2 Conservation

We assess the conservation property of schemes using two parameters to determine whether there is a loss or increase in sea ice area during transport. The first parameter is the ratio of the total ice area after transport to the initial total ice area. The second parameter is the ratio of the ice area that reaches the target area to the theoretical solution, which also indicates the degree of accuracy. In Table 2 we can see that, for the total area of ice, there is no change in area during the transport process using the upwind and

TVD schemes. In contrast, the ice area of the central difference scheme declines initially, and then

increases over time. This suggests that the limiting procedure used in the central difference scheme destroys conservation. The upwind scheme performs relatively well at first, with ~72.34% ice reaching the target area after 1 hour. But its performance drops significantly, to ~25.76% at the end. The central difference scheme, achieves a percentage of ~60.29% at first, and at the end it manages to achieve ~55.11%. Among the three schemes, TVD is the most effective with a consistently higher percentage compared to the others. The percentage of the region reaching the target area exceeds 90% after 1 hour and consistently maintains a rate of nearly 80% towards the end.

|  |  | 1 hour | 3 hours | 6 hours | 12 hours | 24 hours |
|---|---|---|---|---|---|---|
| Upwind | All area | 100.00% | 100% | 100% | 100% | 100% |
|  | In the target area | 79.41% | 65.71% | 53.62% | 39.40% | 25.76% |
| Central difference | All area | 72.34% | 77.85% | 80.92% | 85.50% | 91.82% |
|  | In the target area | 60.29% | 59.08% | 58.88% | 57.08% | 55.11% |
| TVD | All area | 100.00% | 100% | 100% | 100% | 100% |
|  | In the target area | 91.23% | 88.12% | 85.67% | 83.14% | 79.22% |

**Table2.** Ice area as a percentage of the exact solution.

### 3.1.3 Monotonicity

Here we select the ice thickness as the representative of traces to verify monotonicity, with the ideal transport scheme expected to maintain the initial ice thickness (1.5m). Fig.2e and Fig.3e have shown the central difference is not monotone as the ice thickness per unit area exceeds the initial maximum even in the first snapshot (3 hours after the start of the case); so we exclude it in the current comparison. Given that the non-monotonicity usually happens at low concentration areas, we choose 0.1% as the threshold. The upwind scheme is completely monotone and the ice thickness remains consistent with the initial value (Fig. 4a). For the TVD scheme, some overshoots would occur in the forward edge of ice (Fig.4b) if we did not limit the up-upwind value (cf. Eq. 9). On the other hand, the new TVD scheme we developed that limits the up-upwind value (cf. Eq. (10)) is completely monotone (Fig. 4c).



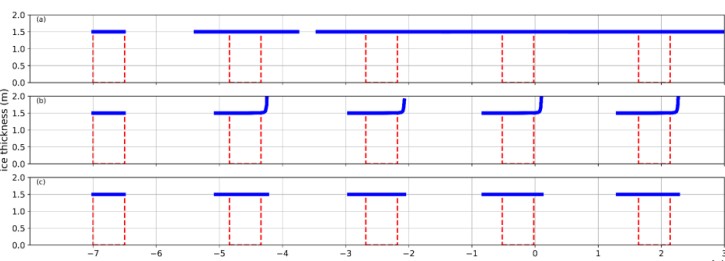

**Figure 4.** Sea ice thickness calculated from **(a)** Upwind, **(b)** original TVD (Eq. (9)), **(c)** new TVD (Eq. (10)) for ice. The time interval of snapshots is every 6 hours.

### 3.2 Realistic model run

A realistic ocean-ice coupled model using the TVD scheme for the ice module is developed to reproduce the ice processes in Lake Superior and the Arctic Ocean. In contrast to the idealized case, both meshes are non-uniform (Fig.5), so the successful tests demonstrate the cross-scale capability of the coupled model.

### 3.2.1 Test on a very high-resolution mesh

To gauge the numerical efficiency of the new TVD scheme, we test it on a very fine resolution Lake Superior mesh (Fig.5a) that was previously used in Zhang et al. (2023). The nearshore resolution in this mesh reaches ~50m with the finest resolution of 41.5m, found on the southwestern shore. As Zhang et al. (2023) indicated, the FCT scheme was having stability issues, so an essentially upwind method was applied in the high-resolution areas. The performance is compared with the upwind scheme. We

simulate the case for 180 days from December 1$^{st}$, 2017, using 48 processors. The total simulation times are similar with 2 schemes, 637 minutes for the TVD scheme and 654 minutes for the upwind. Note that the upwind scheme would be cheaper, but the total times shown include other modules in the model; the diffusion in the upwind scheme has led to a larger ice coverage area, thus increasing the cost of the ice solver. Overall, we found the cost for the TVD scheme is comparable to the upwind scheme

for many realistic applications.

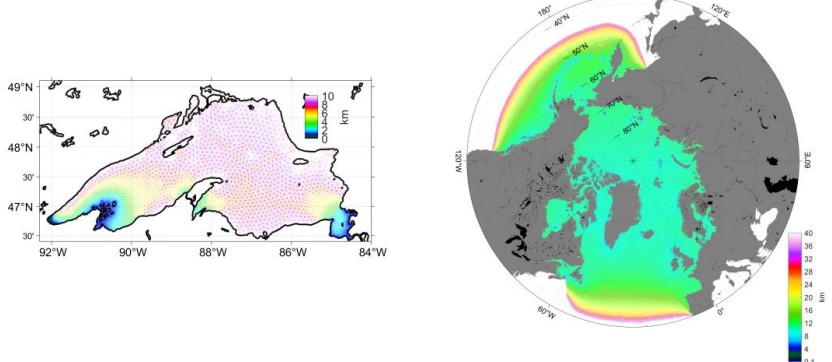

**Figure 5. (a, left)** The Lake Superior mesh. **(b, right)** The Arctic Ocean mesh. The colors show the mesh resolution.

Next, we qualitatively compare results from the two schemes on Day 100 (Fig.6), when the ice cover is

largest. Results of ice concentration are similar, with higher concentration in the nearshore area and

lower concentration in the center of the lake, which is consistent with the single-class ice model of

Zhang et al. (2023). The TVD results reveal more variability especially in the open water. The results

for the ice thickness are quite different, with thick ice further from the coast from the upwind result

than from the TVD scheme (Fig. 6). This is because the upwind scheme is more diffusive than TVD. In

the very high-resolution areas (southwestern and southeastern corners), both schemes yield reasonable

and stable results, which demonstrates the coupled model's cross-scale capability.

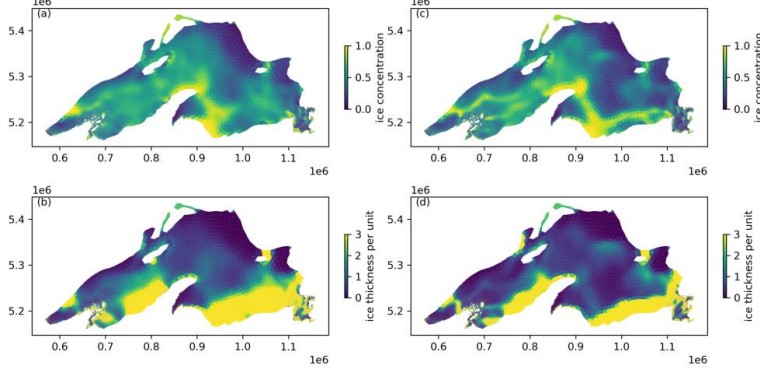

**Figure 6. The** top row is the ice concentration at Day 100, and **(a)** is the result of the upwind scheme **(c)** is that of
the TVD scheme. The bottom row is the ice thickness per unit and **(b)** is for upwind, **(d) is** for TVD.



**3.2.2 Test on the Arctic Ocean**

The Arctic mesh consists of 422,000 elements and 217,000 nodes (Fig. 5b) with the resolution ranging from 6 km near the coast to 40 km at the open boundary. The model starts on January $1^{st}$, 1994, and covers 2000 days, about 1.6 million steps using a time step of 100 sec. The initial condition is obtained from HYCOM, including ocean tracers, sea ice concentration and thickness. Moreover, the boundary condition is obtained from HYCOM and Finite Element Solution (FES2014, Lyard et al., 2021), including 15 tidal components. The domain boundary is chosen to be at ~40°N to ensure no sea ice crosses the boundary. In the vertical dimension, a highly flexible vertical gridding system ($LSC^2$, Zhang et al., 2015) has been implemented with up to 60 layers in order to describe the complex topography of the Arctic Basin better, and we set the bottom drag coefficient with a constant manning coefficient at 0.025. For the atmosphere forcing we choose The European Centre for Medium-Range Weather Forecasts Reanalysis Fifth Generation global reanalysis (ERA5, Hersbach et al., 2020) for its high temporal resolution, and use the bulk aerodynamic model (Zeng et al., 1998) to get the surface fluxes, like latent and sensible fluxes. The turbulence closure scheme in the hydro model is the generic length-scale equation as k-kl (Umlauf and Burchard, 2003) and the horizontal transport in the hydro model is $TVD^2$ (Ye et al. 2016). The parameters used in the sea ice model basically follow the standard CICE configuration, including a constant air-ice drag coefficient (about 0.0016), and a constant ice-ocean drag coefficient (about 0.006). Modules in the standard CICE are also included in this model, e.g., the mushy layer thermodynamics, the Rothrock (1975) ice strength method, the level-ice melt ponds module, etc. We compare the results of our Arctic sea ice model with the NSIDC observation, including the sea ice extent, ice boundary, and ice concentration. The observation of Sea Ice Concentrations is from Nimbus-7 SMMR and DMSP SSM/I-SSMIS Passive Microwave Data (Fetterer et al., 2017), while the sea ice boundary corresponds to the 15% sea-ice concentration contour. Fig.7a compares the sea ice extent of our model with the observation. The model is stable for the long-term test and has good performance to reproduce the inter-annual variability and the seasonal cycle, with the minimum and maximum being reproduced satisfactorily. The first peak is higher than the observed value, which may be influenced by the initial conditions as we did not get all tracers, such as sea ice salinity and enthalpy, from HYCOM. The extent difference between the model and observation is evaluated as absolute extent error (AEE, Eq. 15). However, AEE may underestimate the model error





due to the cancellation between the overestimation(O) and underestimation(U). The Integrated Ice

Edge Error (IIEE, Eq. 16) may be a preferable choice to evaluate the simulation result (Goessling et al., 2016, Zampieri et al., 2018).

$$AEE = |\sum(|O| - |U|)|, \tag{15}$$

$$IIEE = \sum|O| + \sum|U|, \tag{16}$$

We present the monthly AEE and IIEE in Fig. 7b, provide monthly statistics for them and compare our

results with the FESOM2's in Fig. 7c. FESOM2 team has run multiple cases to investigate the sensitive of results to various forcing and model complexities, and the ones we selected were also driven by ERA5 and based on a multi-class ice thermodynamics BL99 (Zampieri et al., 2021). IIEE and AEE (Fig. 7b) fluctuate in a similar fashion to the monthly extent in Fig.7a. In Fig.7a, the simulated sea ice extent often increases faster in autumn than observation, and it seems to perform better in other seasons. AEE shows

a similar pattern, being relatively small in spring and summer, and reaching its maximum in autumn. The magnitude of AEE is also similar to that of FESOM2, peaking in autumn while lower in other seasons. The seasonal pattern of IIEE is similar to that of FESOM2, with maximum values during summer and lower values during autumn. The largest variability of IIEE occurs in summer, too, while the lowest variability is observed in spring. The differences between our model and FESOM2 may be caused by

two factors. The first is that the results we select to compare from FESOM2 use different thermodynamic module and the second is that the integration period of FESOM2 is from 2002 to 2015, while this study's integration period is from 1994 to 1999. Nonetheless, the performance of the two models seems generally comparable.

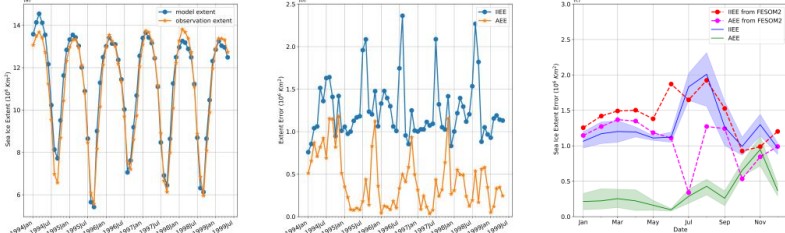

**Figure 7. (a)** Monthly sea ice extent of model and observation in the Arctic Ocean. **(b)** Monthly Integrated Ice Edge Error (IIEE) and Absolute Extent Error (AEE). **(c)** Monthly IIEE and AEE of our model and FESOM2 (averaged across all years), with the shading representing the 95% confidence intervals.

The comparison of the spatial sea ice concentration is shown in Fig.8. The simulated sea ice boundary and ice concentration show good agreement with the observation, and the model shows a robust ability



to capture the seasonal evolution of sea ice in the Arctic Ocean. During winter and spring (Fig. 8a and

8b), the deviation occurs in the marginal ice zone, such as the Bering Sea and the Atlantic Ocean. In

summer (Fig.8c), the model overestimates the sea ice concentration near the coast, such as the

Canadian archipelago coast, but underestimates in the central Arctic Basin. The overestimation is likely

due to the presence of complex thermodynamic and dynamic processes in the coastal margin (e.g., the

occurrence of landfast sea ice). Furthermore, the lack of precise runoff and temperature data of Arctic

rivers has a significant impact on the coastal area simulation. In the central Arctic Basin, melt ponds

have a significant effect on the mass of sea ice during the melting season, and they are always formed

as a certain amount of precipitation remains on the ice (Feng et al., 2022). The precipitation field we

utilized shows some overestimation compared to the observation-based precipitation products

(Marcovecchio et al., 2021), so the underestimation of sea ice concentration in the central Arctic Basin

is likely due to the excessive melt ponds that were reproduced in our model. In autumn (Fig.8d), the

model overestimates sea ice concentration in the marginal seas of the Arctic, like Hudson and Baffin

Bay, which causes the largest AEE. The heat exchange between the air-ice-sea interface is generally

more intense in the freezing season than melting season, so the coupled model may generate more sea

ice due to its inability to deliver sufficient heat to the surface in time or due to the insufficient strength

of convection in the upper ocean.

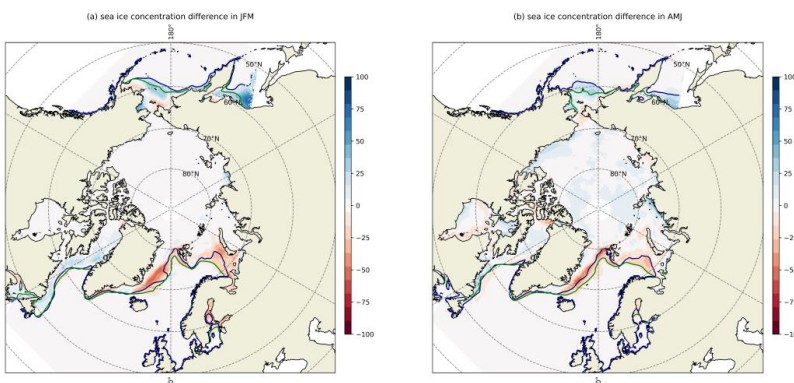



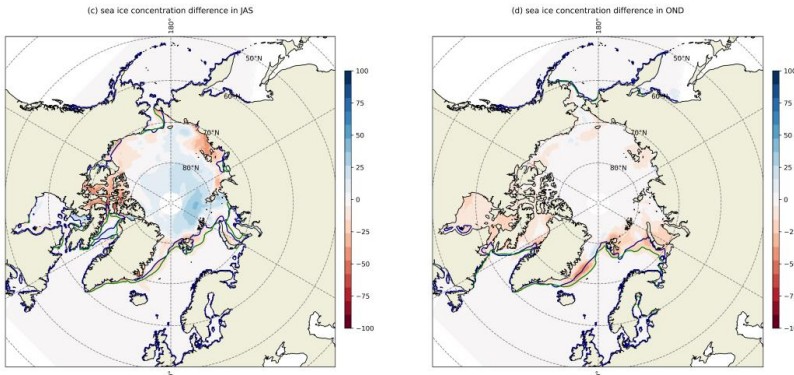

**Figure 8.** Seasonally averaged sea ice concentration difference (Observation-Model). The blue line is the satellite sea ice boundary, and the green line is from the model. **(a)** Winter (Jan. Feb. and Mar.), **(b)** Spring (Apr. May. and Jun.), **(c)** Summer (Jul. Aug. and Sept.), **(d)** Autumn (Oct. Nov. and Dec.)

## 4 Discussion

In this study, the grid definition is different between the ice module and the hydrodynamic module. The ice module uses the Arakawa-A grid, and all tracers and velocities are defined at nodes, while the hydrodynamic module uses the Arakawa-CD grid. The main reason behind this decision is that we adapted the rheology part from FESIM, which uses an analogue of the Arakawa-A grid, and it performs well for sea ice simulation while saving computational costs (Danilov et al., 2015). For a coarse mesh, the Arakawa-A grid delivers performance similar to that of the Arakawa-CD grid in simulating sea ice deformations (Mehlmann et al., 2021). In a pure advection case, the oscillations are weaker for the Arakawa-A grid with tracers located at the node, compared to the case with the tracers located at the centroid (Zhang et al., 2016). The coupling between the ice module and the hydrodynamic module remains unaffected by the differences in the variable definition, as all forcing variables are located at nodes in the hydrodynamic module. Under the uniform mesh in the idealized test, the central difference scheme is similar to the second order upstream scheme of UG-CICE, and the latter achieved remarkable results (Gao et al., 2013). The difference in stencils used between UG-CICE and this work may explain why the central difference scheme has an unsatisfactory performance: in UG-CICE, tracers are also at vertices (nodes) but the velocity is at the centroids. Still, it remains unclear if UG-CICE is strictly monotone.





The TVD scheme used in this work is based on the gradient of the central node, whereas Casulli et al.

(2005) used the flux into the element to obtain the $\phi_{U_*}$ in order to avoid unphysical

overshoots/undershoots. For the Casulli's TVD scheme, the tracers are always located at the centroid,

but they also can be converted to the node (Zhang et al., 2016). A comparison of the results for the

idealized cases from Casulli's TVD and our TVD scheme revealed that Casulli's TVD scheme has more

diffusion than the new TVD scheme.

**5 Conclusion**

We have incorporated a multi-class sea ice module, the advanced sea ice column physics package Icepack,

into the SCHISM modelling system. Significantly, we have implemented a new TVD based scheme for

ice tracer transport and validated it using an idealized case and realistic cases. The simulation results

demonstrate that the TVD scheme is conservative, accurate, strictly monotonic, and efficient in

reproducing the horizontal transport of ice and has better performance than the upwind and central

schemes. The coupled model for the Arctic Ocean was able to reproduce the Arctic Sea ice concentration,

boundary, and extent as seen from the observation.

An advantage of the coupled SCHISM-Icepack is its ability to effectively simulate ocean-ice evolution

in both open ocean and coastal regions. In addition, SCHISM includes various biogeochemistry modules

like CoSiNE, while Icepack contains a biogeochemistry module as well. We will further investigate the

under-ice ecosystem changes caused by global warming by integrating those biogeochemistry modules.

**Code and data availability.**

Code of this model have two components, Icepack 1.3.4 and SCHISM v5.11. Icepack 1.3.4 is obtained

from https://github.com/CICE-Consortium/Icepack . SCHISM v5.11 and the coupled model can be found

at https://github.com/schism-dev/schism, including all the code used in this paper. All source code is

also available on Zenodo (https://doi.org/10.5281/zenodo.10391035, Wang et al., 2023) with all

configuration files of the idealized case and the realistic test on the Arctic Ocean. In the realistic test on

the Arctic Ocean, the forcing data is from ERA5, initial and boundary data is from HYCOM and



FES2014, they can be generated by the preprocessing script in SCHISM. The input data of the realistic

case on the Lake Superior is available from Y. Joseph Zhang on reasonable request. All the results in

the paper are also available from Qian Wang on reasonable request.


## Author contributions

**Qian Wang**: Data curation, Formal analysis, Investigation, Methodology, Software, Visualization,
Writing – original draft preparation. **Fei Chai**: Conceptualization, Supervision, Funding acquisition,
Project administration. **Yang Zhang**: Validation, Resources, Software, Writing – review & editing. **Y.**
**Joseph Zhang**: Validation, Methodology, Software, Writing – review & editing. **Lorenzo Zamperi**:
Writing – review & editing.

## Competing interests

The authors declare that they have no conflict of interest.

## Acknowledgements

The authors acknowledge the financial support of the National Natural Science Foundation of China
(Grant Number 41941013). The study in this paper was supported by the high-performance computing
clusters at: (1) State Key Laboratory of Satellite Ocean Environment Dynamics, SIO, MNR; (2) William
& Mary Research Computing (URL: https://www.wm.edu/it/rc). The authors also thank HYCOM data
server for their sea ice product. Lorenzo Zampieri acknowledges the financial support of the Italian
National Recovery and Resilience Plan (PNRR) through the "SPOKE 4 - EARTH & CLIMATE"
program.

## Financial support

This work is supported by the National Natural Science Foundation of China (Grant Number 41941013)

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
