# Peer review of "Development of a total variation diminishing (TVD) sea ice transport scheme and its application in an ocean (SCHISM v5.11) and sea ice (Icepack v1.3.4) coupled model on unstructured grids"

_Geoscientific Model Development, 2023_

## Referee Comment (RC2)

Review of gmd-2023-236

**General comments**

Wang et al. describe a new version of the SCHISM hydrodynamic model. They have implemented a total-variation-diminishing (TVD) scheme for ice concentration and tracers, and they have coupled SCHISM to a multi-category column package, Icepack, from the CICE model. They show that the TVD scheme meets model requirements (accuracy, conservation, monotonicity, and efficiency) and generally performs well. They also present results from a coupled multiyear ice–ocean simulation of the Arctic region, showing good agreement with observations.

My main issue is that the paper seems to have two distinct aims—analyzing the TVD scheme and validating the overall model—without doing either in an optimal way. The TVD analysis includes a detailed comparison to upwind and centered schemes, with results that will be unsurprising to anyone familiar with transport schemes. It would be better to compare the TVD scheme to a more sophisticated scheme or schemes (e.g., another second-order monotone scheme), showing that it is either more accurate and better behaved numerically, or able to give similar results at lower computational cost.

The model validation for the Arctic is confined to Section 3.2.2. The results seem promising, but the analysis is short and is limited to ice extent. There is no analysis of the ice thickness simulation, nor is there a comparison to previous SCHISM versions without the TVD scheme and multi-category physics. This makes it hard to assess what has been achieved with the new version.

The paper could be much stronger if it were reorganized, with some material dropped and new material added. For the TVD analysis, most of Section 3.1 could be cut. For the overall model validation, Section 3.2.2 could be expanded. The Introduction could do a better job of motivating the model upgrades, and the Conclusion could give a more complete summary of what has improved and what work remains for the future.

Specific comments and corrections follow.

**Specific comments**

l. 15    "A more advanced sea ice transport scheme is needed." The authors give no evidence for this. Rather, the need seems to be for a conservative, monotonic, efficient transport scheme for SCHISM in particular.

l. 18    "Compared with the upwind scheme and a central difference scheme." This seems like a straw-man comparison; it is not at all surprising that a TVD scheme would outperform these two schemes. More on this below.

l. 35    For sea ice ridging processes in CICE, I suggest citing Lipscomb et al. (2007, JGR) rather than Hunke (2010). Also at l. 111.

ll. 35ff The list of models using CICE or Icepack seems secondary to the main point, and a bit random. It is unclear if or how these various models (e.g., UG-CICE and FESOM2) are related to SCHISM. I suggest first describing SCHISM, the kinds of problems it is used for, the previous implementation of sea ice in SCHISM, and the science goals that explain the need for new and improved components.

l. 59    "sea ice coupled models." Does coupling refer to GCMs and ESMs, or just coupling to ocean models?

l. 61    This is the first use of the term "monotonic" in the main text. Here I suggest defining monotonicity in the context of sea ice transport. Typically, this term refers to schemes that don't introduce spurious new maxes or mins in tracers such as ice thickness or enthalpy. For the case of ice concentration, it would refer to schemes that allow new maxes or mins only when the velocity field is convergent or divergent.

l. 63    "Lipscomb et al. (2004)."  Should be "Lipscomb and Hunke (2004)"

l. 67    I don't understand the claim that incremental remapping (IR) is inefficient for unstructured grids. The geometric part of the IR computation scales linearly with the number of grid cells, and the tracer-reconstruction part scales superlinearly with the number of tracers. For CICE and MPAS-Seaice users, the cost of transport is typically not greater than the cost of EVP dynamics and Icepack column physics. If the SCHISM developers opted for TVD in favor of IR, it likely wasn't for reasons of computational efficiency alone. Perhaps they wanted a scheme that was easier to code?

p. 76    "It is unclear…". I don't know why it would be unclear whether or not a scheme is monotonic.

l. 80    For many transport schemes (IR is an exception), the cost increases linearly with the number of variables. Likewise, there is always some cost (usually justified) to imposing strict monotonicity. I'm not sure why these were reasons to rule out FEM-FCT.

l. 81    Here the authors describe the simplicity of the previous version of SCHISM: upwind transport, 0-layer thermodynamics, etc.  They could expand on this discussion to say why there was a need for Icepack and other upgrades.

l. 89    "The performance of the multi-class sea ice formulation has not been tested before."  This suggests the value of a more complete validation as suggested above.

l. 92    This is where TVD is introduced as a scheme with the desired properties. Can the authors define the method and say when and by whom it was introduced? Does it have a prior history in sea ice modeling?

l. 106   The ITD implementation in Icepack is based on Lipscomb (2001), not Bitz et al. (2001).

l. 120   What is meant by "hydrodynamic core"? Is this the ocean model, or is it something more general than an ocean model?

l. 137   I'm not sure it's accurate to say that transport is "the main challenge." Thermodynamics and ridging are challenging too.

p. 141  "a strictly monotone scheme is still desirable".  See the l. 61 comment; it would be better to define and discuss monotonicity earlier.

l. 143   It would be better to introduce the SCHISM model and grid earlier. Please say what is meant by an Arakawa CD-grid.  Does the first use of "SCHISM" on l. 144 refer to the SCHISM lake/ocean component?

l. 146   "centroids".  Meaning centroids of triangles, as opposed to centroids of hexagons?

l. 163   It might be helpful to give the reader some examples of Eqs. (7) and (8) in action, showing how they work to preserve monotonicity. For example, one could consider the three cases of ($phi_C$ – $phi_U^*$) = ($phi_D$ – $phi_C$), 0, and  -($phi_D$ – $phi_C$), which would imply $psi_i$ = 1, 0, and 0, respectively.

l. 169   "gradient of the central node grad($phi_C$)".  Does this mean the quantity grad($phi$), evaluated at node C? How is the gradient evaluated? E.g., with a line integral around the adjacent nodes?

l. 181   I'm not sure $phi_U^*$ is meant here, since it's not an edge tracer value. Should this be $phi_i$, as computed in Eq. 6?

l. 181   The term "sea ice fluxes" is ambiguous. Does this mean fluxes of ice area?

l. 182   Is van Leer limiting applied to $h$ and $q$?  I think it must be, if $h$ and $q$ are to be advected monotonically.

l. 205   "Since the thermodynamic part…".  Icepack is a significant step forward for SCHISM, so it would be interesting to know if it improves results compared to earlier model versions, for either the Great Lakes or the Arctic.

l. 211. A time step of 1 s seems unnecessarily short given the size of the triangles (200 m on a side) and speed of the flow (1 m/s).

l. 213 It is predictable that TVD will outperform upwind and centered difference schemes in exactly the ways described. There is no need to include conservation as a metric, since TVD (like upwind) is conservative by construction, whereas centered is not (given that over- and undershoots are clipped). Thus, the following three sections (on accuracy, conservation, and monotonicity) are longer than necessary and not very illuminating.

A more relevant analysis would be to compare TVD to incremental remapping (if the authors were able to set up similar test problems in CICE or MPAS-Seaice) or another second-order monotone scheme. In the case of IR, it could be interesting to show that TVD gives similar results at lower cost.

l. 289 The high-resolution Great Lakes simulation is a good problem for comparing TVD and upwind. The results shown in Fig. 6 are quite convincing. For this reason, I think the authors could leave out the simple problems in Section 3.1 and let Section 3.2.1 make the case for TVD over upwind.

It is interesting that the TVD method reduces the overall model cost (compared to upwind) by limiting diffusion of ice area. How much time is spent in the transport solver alone for each of the two transport schemes, and how does this compare to the total model time?

l. 304 I suggest "compare" instead of "qualitatively compare", since the comparison is not merely qualitative.

l. 315 Since this section focuses on general model validation (rather than a validation of TVD), I suggest expanding it and making it an entire section rather than a subsection. Also, it would be useful to see how the new model version compares with the older, simpler version.

l. 318 Is the sea ice time step just 100 s? This seems unnecessarily short if the minimum grid cell size is 6 km, assuming a max speed of ~1 m/s.

l. 328 I am not sure what is meant by "the generic length-scale equation as k-kl."

l. 330 How is TVD$^2$ related to the TVD scheme implemented for the sea ice model?

l. 341 "which may be influenced by the initial conditions as we did not get all tracers, such as sea ice salinity and enthalpy, from HYCOM." I can think of many reasons why the first peak might not line up with the observed value. I'm not sure why initial tracer values are singled out as an explanation.

l. 350  Why was FESOM2, as opposed to some other model, chosen as a standard for comparison? How similar was the FESOM2 configuration?

l. 354  "the simulated sea ice extent often increases faster in autumn than observation." This isn't obvious from Fig. 7a. I just see one year (1994) when the modeled September min is significantly greater than observed.

l. 361  Typically when comparing two models, one would force them over the same integration period. If FESOM runs are available from 1994–1999, I would suggest using those. If not, then it might be better to leave out the comparison.

l. 368  It's helpful to see these spatial patterns of sea ice concentration biases. Would it be possible also to show plots of sea ice thickness compared to observations?

l. 371  Do the authors know why the ice edge is too far advanced on the Atlantic side, and not far enough on the Pacific side? Is this likely an ocean model bias?

l. 381  The melt pond hypothesis is interesting. Is it possible to test this idea by, for instance, turning off melt ponds or using different precipitation forcing?

l. 393  The discussion section is short and includes some material (e.g., grid choice) that would fit better earlier in the paper. It doesn't shed new light on the Section 3 results. I would suggest leaving it out.

l. 405  I doubt that "remarkable" is the right word here. Again, I don't think the centered difference scheme adds value to the Section 3 analysis.

l. 410  This is an odd place to introduce the Casulli et al. scheme. Maybe do this earlier, in Section 2.2 or 3.1.

l. 417  The conclusion is short and cursory. It would be better to include a discussion of how the addition of Icepack and the TVD scheme have improved SCHISM compared to the previous model version.

l. 460  The reference list is incomplete and contains some errors. For instance, there is no Gurvan et al. (2022) or Campin et al. (2023).

**Minor corrections**

l. 23    "the satellite" -> "satellites"
l. 25    "dramatically" -> "dramatic", "Sea ice" -> "sea ice"
l. 29    "the sea ice models" -> "sea ice models"
l. 87    "The Great Lake" -> "the Great Lake"

l. 109 (and elsewhere):  "traces" -> "tracers"

l. 132 (and elsewhere):   "Where" -> "where"

l. 139 (and elsewhere):  Check punctuation with equations. Here, the comma should be a period.

l. 162   "Van-leer" -> "van Leer".  Also l. 197.

l. 183   "as does CICE" -> "as in CICE"

l. 324   "manning" -> "Manning"

l. 325   "The" -> "the"

l. 335   "Sea Ice Concentrations" -> "sea ice concentration"

This is not a complete list. There are many minor typographical and grammatical errors that should be cleaned up in the next version.

---

## Author Response (AR3)

Reply to referee 1

The manuscript presents some interesting developments for SCHISM and may be of interest for the respective community, and possibly beyond. I recommend a revision along the following lines.

General answer:

Thank you for your comments and we appreciate the time you have taken to review this manuscript. We have revised the manuscript according to your suggestions and will respond to your comments paragraph by paragraph.

I do not see that the introductory part gives a relevant story, which is: There is a certain model (SCISM) that already has a sea-ice module. This module is further improved by adding the ICEPACK and by adopting the advection scheme proposed in the manuscript. While the need to include ICEPACK does not require much motivation, the need to develop a new advection scheme is not properly explained. In addition to the advection schemes used by the available UG sea-ice models, numerous other advection schemes (including the TVD schemes) have been proposed in the literature for unstructured meshes. In the manuscript, only one is mentioned outside the models, and only at the end (by Casulli). An analysis of the existing approaches is needed, explaining what was missing and why the authors choose to develop the new approach.

Answer: Thank you for your comments, and we have reorganized the Introduction as you suggested. The motivation have been discussed firstly in the revised version, and has been summarized below. Before this study, the released version of SCHISM includes a simple single-class ice model, and Zhang et al. (2023) had used it to simulate the Lake Superior ice, while the simulation results showed the ice melting is faster than observation in the model. The previous simple single-class ice model in SCHISM lacked the sophisticated features present in Icepack, such as ice thickness distribution, melting ponds and ice ridges. Our motivation for implementing Icepack is to improve the accuracy of the ice results. When attempting to implement the multi-class ice model, the previous transport scheme, FEM-FCT, showed signs of instability. In the Lake Superior case, Zhang et al. (2023) also modified it by zeroing out the higher-order contribution to keep stability. Given these instabilities and the potential for improved performance, we were motivated to develop a new transport scheme that could offer greater stability and accuracy, which hopefully can provide guidance or serves as a reference for future similar works. After comparing several transport schemes, including upwind, central difference, and second-order upwind scheme, the TVD transport scheme was selected. We've seen promising results with the TVD scheme in SCHISM's application to ocean tracers, where it has successfully managed the salinity, temperature, and other tracers.

It should be noted that the TVD in SCHISM is based on the flux into the element (by Casulli), the upwind ratio $r_i$ is derived from the neighbouring elements:

$$r_i = \frac{1}{\phi_C - \phi_D} \times \frac{\sum_{m \in S^-} |Q_m|(\phi_m - \phi_C)}{\sum_{m \in S^-} |Q_m|},$$

$S^-$ is every inflow faces of the control volume, $Q_m$ is the flux across the edge, other values are same as them in manuscript. And for the sea ice model, it is based on the gradient of the central node (by Darwish, Eq.7 in manuscript), these two schemes are similar. So actually, there is no schemes outside the models.

Secondly, given that the other focus of the manuscript is on the implementation of ICEPACK, I expected to see a demonstration of the changes brought about by ICEPACK, compared to the previous version of the sea ice model. I do not find such a discussion in the manuscript. I also miss the discussion of the computational aspect of the ICEPACK implementation, as well as on the computational aspect of running SCHISM in the Arctic Ocean, which is a remarkable undertaking in itself.

**Answer:** We value your feedback and recognize the importance of elucidating the advancements facilitated by integrating ICEPACK. To this end, we have employed the SCHISM-Icepack model to revisit the case study of Lake Superior ice (Zhang et al., 2023), enabling a direct comparison against the preceding single-class ice model.

Regrettably, due to computational resource and time constraints, we did not conduct extensive sensitivity tests in the Arctic Ocean case. But We do include a discussion on the computational efficiency of the transport schemes used in the Lake Superior case.

These comparisons are added in section 3.2.1 and also shown below. The comparison between the new version and the released version of SCHISM is in the first and the second paragraphs below, while the third paragraph below is about the computational efficiency of the transport schemes.

As we have mentioned before, Zhang et al. (2023) have used a single-class ice model to reproduce the seasonal and interannual variability of ice extent (the ice concentration greater than 15%) in this case. The simulation results have been compared to the Great Lakes Surface Environmental Analysis (GLSEA) data, including some rapid melting-refreezing events. But they also found in their model that ice melts excessively fast near the end of each melting season. Here we compare the ice extent and ice concentration between two models. With the multi-class ice model and the TVD scheme, we are able to reproduce the similar pattern of ice extent and also some rapid melting-refreezing events, yielding a correlation coefficient of 0.93 and a Wilmot score of 0.92 (Fig. 1). Furthermore, the melting phase simulation is improved beyond day 120. Approximately 10,000 km² of ice persists until around day 150 and the ice dissipates by day 160, aligning more closely with observational data. After the observed ice extent falls below 10,000 km², the correlation coefficient using the multi-class ice model is 0.82, which is an improvement over the single-class ice model's coefficient of 0.43.

[Figure]

**Figure 1.** Comparison of ice extent in Lake Superior in 2017, the blue line is the result of multi-class ice model, the orange line is the result of single-class ice model, and green dot is the observation from GLSEA.

The spatial distribution of the ice concentration from the two models is compared with the observation from the U.S. National Ice Center (USNIC) on day 90 (Fig.2), when the ice cover was largest. Both models exhibit lower ice concentration in the southern part of the lake; however, while in most other

areas, particularly in the western region, the multi-class ice model displays lower ice concentrations. Compared to the USNIC data, both models overestimate ice concentration on the lake's eastern side. However, the multi-class ice model reproduces the ice-free pattern on the west coast more accurately. The spatially average ice concentration is 0.617 for the multi-class ice model, which is closer to the observed value of 0.509 and represents a significant improvement over the single-class ice model's 0.847. In the very high-resolution areas (the lake's southwestern and southeastern corners) , the new coupled model and the TVD scheme yield a reasonable and stable result, which demonstrates its cross-scale capability.

[Figure]

**Figure 2.** The ice concentration on day 90, and **(a)** is the result of the multi-lass ice model **(b)** is that of single-class ice model, **(c)** is from USNIC.

The performance is compared with the upwind scheme. We simulate the case for 180 days from December 1st, 2017, using 60 processors. The total simulation times for the two schemes are comparable, 678 minutes for the TVD scheme and 675 minutes for the upwind scheme. In total, the upwind scheme consumes 52.39 core hours while TVD spends 54.56 core hours. Compared to the total time of the ice module, TVD accounts for 21.71% and upwind accounts for 21.01%, while the dynamic part is the most computationally intensive, accounting for more than 70%. Overall, we found that the computational cost of the TVD scheme is comparable to that of the upwind scheme in this realistic benchmark application.

Thirdly, much of the material in the discussion of the performance of the new advection scheme is on an almost trivial level: the issues accompanying the first-order upwind or central schemes are well known. There is no point in section 3.1.2, since the finite volume methods are by definition conservative, and the authors have already mentioned this in the manuscript. The 'non-conservation' discussed here is due to clipping for the central differences. No one will use central differences when positivity is required. This section needs to be removed. Compare the new scheme with those used by other models or known from

the literature and discuss its advantages. This will give a useful message to a wider community of modellers. Otherwise --- almost any sophisticated scheme will outperform the two used here for the comparison. Therefore, the analysis in 3.1 lacks substance.

**Answer:** In the initially conceived Section 3.1, our objective was to validate that our proposed TVD scheme is not only accurate and stable but also conserving quantities, maintaining monotonicity, and is computationally efficient. Drawing inspiration from Lipscomb and Hunke (2004) and Turner et al. (2022), who juxtaposed the incremental remapping scheme with the traditional upwind method across divergent mesh frameworks, our prior submission presented comparative analyses utilizing a first-order upwind scheme and a CD scheme that is not inherently conservative. We agree that the issues accompanying the first-order upwind or central schemes are well known. As per your feedback, we have revised Section 3.1 substantially and summarized below.

The upwind scheme and CD scheme have been replaced by FEM-FCT and second-order upwind scheme in the idealized case. Our idealized model experiments indicate that while FEM-FCT achieves second-order accuracy (Fig.3) but not monotonic (Fig.4). The second-order upwind scheme is reproduced by UG-CICE, and it does exhibit monotonicity; however, our TVD scheme surpasses it in terms of accuracy (Fig.3). Notably, within UG-CICE, tracers are located at vertices (nodes), and velocity is defined at centroids – a configuration that diverges from our model's approach.

[Figure]

**Figure 3.** Sea ice concentration snapshots **(a-c)** and profiles **(d-f).** The sea ice moves from left to right, snapshots are taken every 3 hours, and the red rectangular is the exact solution. **(a, d)** second-order upwind, **(b, e)** FEM-FCT, **(c, f)** TVD

[Figure]

**Figure 4.** Sea ice thickness calculated from **(a)** FEM-FCT, **(b)** original TVD (Eq. (8) in the manuscript), **(c)** modified TVD (Eq. (9) in the manuscript) for ice. The time interval of snapshots is every 6 hours.

There are many specific points, some minor, but some requiring more attention. These are listed below.

Line 40 and in other places: Gurvan is the name, not the family name. The list of references contains errors, please check and correct (line 485 - non-standard, 527 the list of authors is non-standard and contains a lot of garbage, 544 non-standard).

**A1:** Sorry for the careless, and we have corrected them. I was misled by Zenodo as they put family name first. Line 485, line 527, and 544 have also been corrected.

Line 40 has been revised to '(NEMO, and NEMO can also couple with CICE or The Louvain-La-Neuve sea ice model, LIM3, Madec et al., 2022)' in line 44 of the track-changes file.

Line 485 has been revised to 'M.S. Darwish, F. Moukalled. TVD schemes for unstructured grids. International Journal of Heat and Mass Transfer, 46(4), 599-611. https://doi.org/10.1016/S0017-9310(02)00330-7, 2003.' in line 733 of the track-changes file.

Line 527 has been revised to 'Alistair Adcroft, Jean-Michel Campin, Ed Doddridge, Stephanie Dutkiewicz, Constantinos Evangelinos, David Ferreira, Mick Follows, Gael Forget, Baylor Fox-Kemper, Patrick Heimbach, Chris Hill, Ed Hill, Helen Hill, Oliver Jahn, Jody Klymak, Martin Losch, John Marshall, Guillaume Maze, Matt Mazloff, Dimitris Menemenlis, Andrea Molod, and Jeff Scott. MITgcm user manual. Massachusetts Institute of Technology, https://readthedocs.org/projects/mitgcm/downloads/pdf/latest/, 2023.' in line 626 of the track-changes file.

Line 544 has been revised to 'Madec Gurvan, Romain Bourdallé-Badie, Jérôme Chanut, Emanuela Clementi, Andrew Coward, Christian Ethé, Doroteaciro Iovino, Dan Lea, Claire Lévy, Tomas Lovato, Nicolas Martin, Sébastien Masson, Silvia Mocavero, Clément Rousset, Dave Storkey, Simon Müeller, George Nurser, Mike Bell, Guillaume Samson, Pierre Mathiot, Francesca Meleand Aimie Moulin: NEMO ocean engine, https://doi.org/10.5281/zenodo.6334656, 2022.' in line 725 of the track-changes file.

line 45 You are talking about implementation of ICEPACK, so the reference should be on the implementation.

**A2:** We have changed reference of FESOM2 to Zampieri et al., 2021.

This part has been revised to 'the Finite-volumE Sea ice-Ocean Model version 2 (FESOM2, Zampieri et al., 2021)' in line 50 of the track-changes file.

line 67 'but is ineffcient ...' ??? How do you know? Please provide a reference; line 67 is pure fantasy -- fine meshes can also be used in this case. I do not see that the authors fully know what they are writing about.

A3: We apologize for the unclear statement. In Lipscomb and Hunke (2004) and Turner et al. (2022), they all say that the maximum time step may have to be reduced to ensure that trajectories do not cross. For fine mesh in some complex area, like rivers and fjords, there may be some highly distorted unstructured grid, which leads to frequent cross trajectories, so the sub cycling of transport is needed and it will reduce the efficiency. We have detailly describe this issue instead of using the word 'inefficient' in the revised paper.

This part has been revised to 'The incremental remapping scheme is a second-order accurate scheme, and has great performance in structured grid models, but requires excessively smaller time step to avoid cross trajectories for highly distorted UGs.' in line 94 of the track-changes file.

71 Again 'Gurvan'

Same as A1.

72 'The efficiency ...' These are regular mesh model. Their efficiency has been tested in numerous applications. Nobody is going to apply these models at resolutions going to 10 m. Better remove altogether.

A4: Thanks for the reminder, and we have removed the sentence.

80 'but it cost is linearly increasing' -- but this is the case with all the schemes mentioned above except for incremental remapping, but even in this case there is a significant increase.

A5: For FEM-FCT, the computational cost is twice as much as normal finite element Taylor–Galerkin scheme as they add an anti-diffusive term, and in general, it also needs sub-cycle to satisfy the Courant condition. So when it is used to multi-class ice model, the increase in the computational cost can be substantial. Especially, the smaller the time step leads to greater stability and more monotonic tracers, so in Zhang et al. (2023), to consider stability and efficiency both, they only use the lower solution of FEM-FCT, which is monotonic and similar to upwind scheme.

81 'strict monotonicity comes with a high cost' -- what is meant? FCT is not cheap, but the first order upwind mentioned further is much less accurate.

Answered together in A5.

92 'a new model?'    I would call it an update of the SCISM for ICEPACK and new advection scheme for the sea ice.

A6: Thanks for the suggestion, and we have corrected to 'This paper presents SCHISM-Icepack, an unstructured ice-ocean coupled model that updates SCHISM for Icepack.' in line 127 of the trackchanges file.

93 Strict monotonicity is not very welcome because the sea-ice velocities have non-zero divergence. Tests carried out in the manuscript do not explore the order.

A7: We apologize that we do not clarify the monotonicity. Here it refers to schemes that the new tracers will not be larger or less than the local maximum or minimum, like unphysical ice thickness, ice salinity or temperature. As for ice concentration, firstly, when the sea-ice velocities have non-zero divergence or sea ice moves to coastal area, it can exceed 1 in our transport scheme and leads to ridge which has been described in Icepack. Secondly, in our idealized case, there is no convergence or divergence, so the concentration should be monotonic and the tracers, like thickness, should be equal to the initial value, too. We show that the FEM-FCT do change the ice thickness while the TVD not in the same idealized case in the new version of manuscript.
We have defined the monotonicity in line 82 of the track-changes file as:
For the sea ice model, the monotonicity means the new tracers will not exceed the local maximum or minimum around it (Lipscomb and Hunke, 2004). And for ice concentration, it can exceed 1 and leads to ridge which has been described in Icepack.

132 and many cases below: change 'Where' to 'where'

A8: Thanks for the suggestion, and we have corrected them.

140-142 One needs a positive scheme. The enforcement of monotonicity is an error in the context of sea-ice modeling. The statement made by the authors is strange and throws doubts on the entire paper. Please explain in detail how a monotone scheme can be made appropriate in a converging or diverging velocity field. Note that the first order upwind scheme will is only positive in divergent flows provided time step limitations.

Answered together in A7.

151 approximate

A9: Thanks for the suggestion.

Eq. (10) Is similar limiting applied to thicknesses and enthalpies? Please carefully explain.

A10: In Eq.10, we only applied to ice concentration yet. Thicknesses and enthalpies depend on concentration and it is upwind for these tracers, which mean that the $h_i$ below is the value of the upwind node.

$$v_n^{t+1} = v_n^t + \frac{\Delta t \sum_{i \in S} Q_i \phi_i h_i}{\Omega_S},$$

we will test the limitation to these tracers in further work.

195 The discussion is inconsistent: first strict monotonicity is mentioned, second (200) new extrema are mentioned.

Answered together in A7.

197 The limiter was already mentioned. It is van Leer (not Van-leer)

**A11**: Yes, and we have mentioned it in the Eq.6, here we want to make a small discussion about it. And thanks for the suggestion. We have corrected it.

This part has been revised to 'for which we select the van-Leer limiter (van Leer, 1979),' in line 215 of the track-changes file.

199 Is this reference related to the situation on unstructured grids? There are many flux contributions, and the velocity field is divergent.

**A12**: We are sorry for the unclear statement. The net flux of ice in transport step will induce the concentration exceeding 1, and convergence is a clear example. We have corrected the statement as 'Sea ice concentration can exhibit new extremes after the transport step as a result of physical processes such as convergence and divergence.' in line 260 of the track-changes file.

200-202: Please demonstrate this, the non-negativeness is not obvious from what has been written in the manuscript, and I do not necessarily see how this statement will hold for a divergent flow.

**A13**: We are sorry for the unclear statement. Non-negativeness is talked about the $\phi_{U_*}$ in Eq.10. And the non-negativeness of new value should be guaranteed by small time step. In general, $u \times \Delta t \ll \Delta x$, so $a_n^t \gg \frac{\Delta t \sum_{i \in S} Q_i \phi_i}{\Omega_S}$. But when the scheme is used in very high-resolution case, we also make some limitation, or we can set some sub cycle to reduce the time step in transport module further.

And the monotonicity of tracers is guaranteed because the method in Eq. (12) and Eq. (14) is essentially a weighted average method with non-negative weights. If we consider a divergent flow, the $h_i$ below is just equal to $h_n$, the centre node value.

$v_n = a_n h_n$.

$v_n^{t+1} = v_n^t + \frac{\Delta t \sum_{i \in S} Q_i \phi_i h_i}{\Omega_S}$.

So if we guarantee the $a_n^{t+1}$ is non-negative,

$a_n^{t+1} = a_n^t + \frac{\Delta t \sum_{i \in S} Q_i \phi_i}{\Omega_S}$.

For the divergent flow, here is

$v_n^{t+1} = a_n h_n + \frac{\Delta t \sum_{i \in S} Q_i \phi_i}{\Omega_S} h_n > 0$.

203 Results

**A14**: Thanks for the suggestion and we have corrected it.

220 outline?    shape or form

**A15**: It is the shape. We have corrected to 'The shape of the ice distribution varies over time' in line 287 of the track-changes file.

230 'minor' – they are rather large and quite noticeable.

A16: Sorry, and we describe them more accurately. We show both TVD and FEM-FCT have similar diffusion on the frontal edges in the revised version and the results have been shown in the general answer.

237 The discussion here is too trivial to be included in the manuscript

A17: We compare TVD scheme with FEM-FCT scheme, which is a more sophisticated scheme. Thanks for the suggestion and the results have been shown in the general answer.

Figure 4. I was not able to guess what is shown in this figure. I am also unable to understand what is the intension of this section. Monotonicity for the first order upwind is well known, as well as non-monotonicity for the central differences.

A18: Yes, we intended to show that the modified TVD scheme has same monotonicity for tracers as upwind. In this figure, we select ice thickness as the example, and it is derived by $h_i = \frac{v_i}{a_i}$.

The middle is the original TVD, and the modified TVD has adjustment by $\phi_{U*} = \min\left(1, max\left(0, \phi_D - 2R_{CD} \cdot (\nabla \phi_C)\right)\right)$.

In the revised version, we compare the monotonicity of FEM-FCT and TVD, and find that for FEM-FCT, the thickness overshoots the initial value in the front side and it has oscillation in the tail edge, while TVD is monotonic everywhere. And the results have been shown in the general answer.

286-287 Was there any doubt? Better remove this sentence.

A19: We want to highlight the cross-scale capability. We have modified this sentence as: The successful tests on unstructured grids (Fig.5) demonstrate the cross-scale capability of the coupled model.

299 Why is this related to the increase of cost?

A20: Actually, if there is no ice and no condition to produce ice (surface temperature is above frozen point), less subroutine of Icepack will be called. As upwind is more diffusion, the ice will cover more area, and it will spend more time on thermodynamic module. And in order to clarify the cost of transport, we calculated the cost of transport solve alone and its ratio of the total ice module time. The result is below, and shown in in line 390 of the track-changes file.:
The total simulation times for the two schemes are comparable, 678 minutes for the TVD scheme and 675 minutes for the upwind scheme. In total, the upwind scheme consumes 52.39 core hours while TVD spends 54.56 core hours. Compared to the total time of the ice module, TVD accounts for 21.71% and upwind accounts for 21.01%, while the dynamic part is the most computationally intensive, accounting for more than 70%.

308-309 from the upwind -- in the upwind

than from the TVD -- than in the TVD.

A21: Thanks for the reminder, and we have corrected them.

311. You demonstrate that model works in realistic application, there was no doubt on cross-scale, for Zhang et al. 2023 already showed this.

A22: We want to show that the new transport scheme and the new version of SCHISM also has the cross-scale capability. In Zhang et al. 2023, the ice model is the single-class module and the transport scheme of ice is FEM-FCT. We update them all.

This part has been revised to 'In the very high-resolution areas (the lake's southwestern and southeastern corners) , the new coupled model and the TVD scheme yield a reasonable and stable result, which demonstrates its cross-scale capability.' in line 436 of the track-changes file.

318 The time step is too small for the resolution mentioned.

A23: For general CFL condition, the time step is too small, but in this model, we also add tidal component, so the recommend CFL condition is

$$CFL = \frac{(|u| + \sqrt{gh})\Delta t}{\Delta x}$$

Another reason is that when the horizontal scale is close to vertical scale and dt is also large, there is spurious 'upwelling' in SCHISM as described in SCHISM manual, so we need to reduce the dt.

328 hydro-model    -- sounds strange in this case -- it is ocean model

A24: Thanks for the reminder, and we have corrected them.

This part has been revised to 'The turbulence closure scheme in the ocean model is the generic length-scale equation as k-kl (Umlauf and Burchard, 2003) and the horizontal transport in the ocean model is TVD[2] (Ye et al. 2016).' in line 460 of the track-changes file.

398-339 The CD representation is much more resolving even on coarse meshes (according to the cited paper).

A25: Yes, on coarse mesh, like 8km, the CD is still better than the A, but the gap is much smaller on the fine mesh, like 2km. So when the mesh increase to 10 or 12 km, the mean resolution of our Arctic model, the difference may be less.

400 Which oscillations are meant?

A26: It means that in the region with rapid changes the concentration has an unphysical peak, like 1.1 for ice concentration with Arakawa-C and 1.05 with Arakawa-A.

404 How can the central scheme to be similar? It is not clear what the authors are willing to say here, and furthermore, they are even not certain what the advection scheme in Gao et al. 2013 does precisely. Either remove or explain properly with accurate statements.

A27: In Gao et al. 2013, they call the transport as the second-order upwind scheme, but they do not explain in datil, and they do not demonstrate the monotonicity. We explain it more accurate and do a comparation between second-order upwind and TVD in the revised version. The results have been shown

in the general answer.

410 Casulli's scheme appears all of a sudden

**A28**: We reorganize this part and remove the discussion about different TVD scheme. Because they are not particularly relevant to the work of this article, and we may do more research about it further.

Reference:

Gao, G., Chen, C., Qi, J., and Beardsley, R. C.: An unstructured-grid, finite-volume sea ice model: Development, validation, and application, Journal of Geophysical Research, 116, https://doi.org/10.1029/2010JC006688, 2011.

Great Lakes Surface Environmental Analysis (GLSEA): https://coastwatch.glerl.noaa.gov/satellite-data-products/great-lakes-surface-environmental-analysis-glsea/, , last access: 2 April 2024.

Lipscomb, W. H. and Hunke, E. C.: Modeling sea ice transport using incremental remapping, Mon. Weather Rev., 132, 1341–1354, https://doi.org/10.1175/1520-0493(2004)132<1341:MSITUI>2.0.CO;2, 2004.

The U.S. National Ice Center (USNIC) : https://usicecenter.gov/    last access: 2 April 2024.

Turner, A. K., Lipscomb, W. H., Hunke, E. C., Jacobsen, D. W., Jeffery, N., Engwirda, D., Ringler, T. D., and Wolfe, J. D.: MPAS-Seaice (v1.0.0): sea-ice dynamics on unstructured Voronoi meshes, Geoscientific Model Development, 15, 3721-3751, https://doi.org/10.5194/gmd-15-3721-2022, 2022.

Zampieri, L., Kauker, F., Fröhle, J., Sumata, H., Hunke, E. C., and Goessling, H. F.: Impact of Sea‐Ice Model Complexity on the Performance of an Unstructured‐Mesh Sea‐Ice/Ocean Model under Different Atmospheric Forcings, Journal of Advances in Modeling Earth Systems, 13, https://doi.org/10.1029/2020MS002438, 2021.

Zhang, Y. J., Wu, C., Anderson, J., Danilov, S., Wang, Q., Liu, Y., and Wang, Q.: Lake ice simulation using a 3D unstructured grid model, Ocean Dynamics, 73, 219-230, https://doi.org/10.1007/s10236-023-01549-9, 2023.

Reply to referee 2

**General comments**

Wang et al. describe a new version of the SCHISM hydrodynamic model. They have implemented a total-variation-diminishing (TVD) scheme for ice concentration and tracers, and they have coupled SCHISM to a multi-category column package, Icepack, from the CICE model. They show that the TVD scheme meets model requirements (accuracy, conservation, monotonicity, and efficiency) and generally performs well. They also present results from a coupled multiyear ice–ocean simulation of the Arctic region, showing good agreement with observations.

**General answer**:
Thanks for the comments and we appreciate the time you have taken to review this manuscript. We have revised the manuscript according to your suggestions and will respond to your comments paragraph by paragraph.

My main issue is that the paper seems to have two distinct aims—analyzing the TVD scheme and validating the overall model—without doing either in an optimal way. The TVD analysis includes a detailed comparison to upwind and centered schemes, with results that will be unsurprising to anyone familiar with transport schemes. It would be better to compare the TVD scheme to a more sophisticated scheme or schemes (e.g., another second-order monotone scheme), showing that it is either more accurate and better behaved numerically, or able to give similar results at lower computational cost.

**Answer:** Thank you for your comment. Before this study, the released version of SCHISM includes a simple single-class ice model, the case study of the Lake Superior ice showed the ice melting is faster than the observation in the model (Zhang et al. 2023). Our motivation for implementing Icepack is to improve the accuracy of the ice results. When attempting to implement the multi-class ice model, the previous transport scheme of ice, FEM-FCT, is sometimes unstable. After comparing several transport schemes in other ice model, we decide to implement the TVD transport scheme, which has been implemented in SCHISM for ocean tracers due to its good performance in SCHISM. We have incorporated a comparative analysis between the released version of SCHISM and the ICEPACK-augmented SCHISM in Section 3.2.1, and we have added a validation of ice thickness in Section 3.2.2. The intent of Section 3.1 is to illustrate that the TVD scheme is accurate, stable, conservative, strictly monotonic, and computationally efficient. For comparison, we initially included a first-order upwind scheme and a non-conservative centered difference (CD) scheme. Because in Lipscomb and Hunke (2004) and Turner et al. (2022), they have already compared the incremental remapping scheme with the upwind scheme in difference grids. We agree that it is better to compare with a more sophisticated scheme. The upwind scheme and CD scheme have been replaced by FEM-FCT and the second-order upwind scheme in the idealized case in the revised paper. The results from our idealized model experiments indicate that FEM-FCT is second-order accurate though not monotonic. The second-order upwind scheme,

replicated from UG-CICE, maintains monotonicity; however, our TVD scheme demonstrates superior accuracy. It is important to note that the UG-CICE model places tracers at vertices (nodes) with velocity defined at centroids, which differs from our approach. The comparisons of idealized case have been shown below, and the detailed version could be found in the Section 3.1 of the revised version:

In general, we demonstrate that the TVD scheme provides second-order accuracy and outperforms the second-order upwind in terms of accuracy (FIg.1). The FEM-FCT method has the potential to be more accurate than the TVD scheme (FIg.1); however, its tendency towards non-monotonicity can cause numerical overshoots (Fig.2), consequently leading to unphysical values for salinity or temperature, which might result in model instabilities or 'blowup'. Approaches to solving the non-monotonicity of the FEM-FCT method may result in higher cost (Löhner et al., 1987) or lower accuracy (Zhang et al.,2023). In this test, the TVD scheme not only preserves the tracer monotonicity but also meets other requirements such as accuracy.

[Figure]

**Figure 1.** Sea ice concentration snapshots **(a-c)** and profiles **(d-f).** The sea ice moves from left to right, snapshots are taken every 3 hours, and the red rectangular is the exact solution**. (a, d)** second-order upwind, **(b, e)** FEM-FCT, **(c, f)** TVD

[Figure]

**Figure 2.** Sea ice thickness calculated from **(a)** FEM-FCT, **(b)** original TVD (Eq. (8) in manuscript), **(c)** modified TVD (Eq. (9) in manuscript) for ice. The time interval of snapshots is every 6 hours.

The model validation for the Arctic is confined to Section 3.2.2. The results seem promising, but the analysis is short and is limited to ice extent. There is no analysis of the ice thickness simulation, nor is there a comparison to previous SCHISM versions without the TVD scheme and multi-category physics. This makes it hard to assess what has been achieved with the new version.

**Answer:** We acknowledge the limitations of satellite data availability for our Arctic case model run, which encompasses the years up to 1999. Given that ICESat was not launched until 2004, we primarily have access to data concerning ice concentration and extent from that time. In the revised manuscript, we have incorporated sea ice thickness data obtained from submarines of the SCience ICe Exercise (SCICEX program) provided by the National Snow and Ice Data Center (NSIDC). We have added this comparison to Section 3.2.2, which is also showed below:

The sea ice thickness is also validated. The observed ice thickness data, derived from upward-looking sonar sea ice draft measurements, were collected by submarines of the SCience ICe EXercise (SCICEX, National Snow and Ice Data Center, 1998). The in-situ data are compared with the corresponding model values using a box plot in Fig. 10. The model results closely match the observations in Apr. 1994, Sept. 1997, Aug. 1998, and Apr. 1999, while the bias of the mean thickness is less than 0.6m. Underestimation of the ice thicknesses happens in other months, with the bias of the mean thickness ranging from approximately 1.0m to 1.5m. Specifically in the springs of 1994 and 1999 (April both years and May in 1999), the median thickness exhibits a bias of about 0.6m, which is smaller than over half of the individual CMIP5 models during the same season, where median thickness biases exceed 1.0m (Stroeve et al., 2014). Overall, SCHISM-Icepack demonstrates a robust capability to replicate both the observed seasonal and interannual variability of sea ice thickness.

[Figure]

**Figure 3.** The box plot of sea ice thickness comparation. Cyan is the model result and orange is the data from submarine.

Another comparison is made for the the Lake Superior, to compare the new version ice model and the single-class ice model (Zhang et al. 2023), the results have been added in the revised version and also shown below:

As we have mentioned before, Zhang et al. (2023) have used a single-class ice model to reproduce the seasonal and interannual variability of ice extent (the ice concentration greater than 15%) in this case. The simulation results have been compared to the Great Lakes Surface Environmental Analysis (GLSEA) data, including some rapid melting-refreezing events. But they also found in their model that ice melts excessively fast near the end of each melting season. Here we compare the ice extent and ice concentration between two models. With the multi-class ice model and the TVD scheme, we are able to reproduce the similar pattern of ice extent and also some rapid melting-refreezing events, yielding a correlation coefficient of 0.93 and a Wilmot score of 0.92 (Fig. 6). Furthermore, the melting phase simulation is improved beyond day 120. Approximately 10,000 km² of ice persists until around day 150 and the ice dissipates by day 160, aligning more closely with observational data. After the observed ice extent falls below 10,000 km², the correlation coefficient using the multi-class ice model is 0.82, which is an improvement over the single-class ice model's coefficient of 0.43.

[Figure]

**Figure 4.** Comparison of ice extent in Lake Superior in 2017, the blue line is the result of

multi-class ice model, the orange line is the result of single-class ice model, and green dot is the observation from GLSEA.

The spatial distribution of the ice concentration from the two models is compared with the observation from the U.S. National Ice Center (USNIC) on day 90 (Fig.7), when the ice cover was largest. Both models exhibit lower ice concentration in the southern part of the lake; however, while in most other areas, particularly in the western region, the multi-class ice model displays lower ice concentrations. Compared to the USNIC data, both models overestimate ice concentration on the lake's eastern side. However, the multi-class ice model reproduces the ice-free pattern on the west coast more accurately. The spatially average ice concentration is 0.617 for the multi-class ice model, which is closer to the observed value of 0.509 and represents a significant improvement over the single-class ice model's 0.847.

[Figure]

**Figure 5**. The ice concentration on day 90, and (a) is the result of the multi-lass ice model (b) is that of single-class ice model, (c) is from USNIC.

The paper could be much stronger if it were reorganized, with some material dropped and new material added. For the TVD analysis, most of Section 3.1 could be cut. For the overall model validation, Section 3.2.2 could be expanded. The Introduction could do a better job of motivating the model upgrades, and the Conclusion could give a more complete summary of what has improved and what work remains for the future.

**Answer:** We reorganize the paper and Section 3.1 is modified as the first answer, Section 3.2 is expanded as the second answer. The Introduction has been revised to better articulate the motivation for model improvements, as summarized below.

Before this study, the released version of SCHISM includes a single-class ice model, and Zhang et al. (2023) have used it to simulate the Lake Superior ice, while the simulation results showed the ice melting is faster than observation in the model. Icepack, being a more sophisticated ice model, offers a comprehensive description of ice processes, including melting ponds, ice ridge and so on. Our aim is to enhance the simulation accuracy, thus motivating the implementation of Icepack. And when we try to implement the multi-class ice model, the previous transport scheme of ice, FEM-FCT, is sometimes unstable. In the Lake Superior case, Zhang et al. (2023) also modified it by zeroing out the higher-order contribution to keep stability. So we decide to implement a new transport scheme for the ice which hopefully can provide a reference for future works. Compare several transport schemes in other ice model, and finally we decide to use the TVD transport scheme, which has been implemented in SCHISM for ocean tracers due to its good performance in SCHISM. The Conclusion has been reworked to outline the improvements more explicitly; the TVD scheme's strictly monotonic nature reinforces stability, which is the trouble in the single-class ice model with the FEM-FCT, and the multi-class ice model has better performance than the previous version in the Lake Superior case.

Specific comments and corrections follow.

**Specific comments**

l. 15 "A more advanced sea ice transport scheme is needed." The authors give no evidence for this. Rather, the need seems to be for a conservative, monotonic, efficient transport scheme for SCHISM in particular.

**A1**: SCHISM needs a conservative, monotonic, efficient transport scheme for multi-class ice module indeed, and we have tested many advection schemes of UG and find some shortcomings of them. The details have been shown in the general answer.

l. 18 "Compared with the upwind scheme and a central difference scheme." This seems like a straw-man comparison; it is not at all surprising that a TVD scheme would outperform these two schemes. More on this below.

**A2**: We compare the TVD with some more sophisticate scheme to show the unsuitability of these schemes. The results have been shown in the general answer

l. 35 For sea ice ridging processes in CICE, I suggest citing Lipscomb et al. (2007, JGR) rather than Hunke (2010). Also at l. 111.

**A3**: Thanks for the reminder, and we have corrected them.

These parts have been revised to 'the sea ice ridging processes (Lipscomb et al., 2007)' in l. 39 and 'a mechanical redistribution parameterization (Lipscomb et al. 2007)' in l. 155 of the track-changes file.

ll. 35ff The list of models using CICE or Icepack seems secondary to the main point, and a bit random. It is unclear if or how these various models (e.g., UG-CICE and FESOM2) are related to SCHISM. I suggest first describing SCHISM, the kinds of problems it is used for, the previous implementation of sea ice in SCHISM, and the science goals that explain the need for new and improved components.

**A4**: Thanks for the reminder, and we have reorganized this part and started with SCHISM as summarized in the general answer.

l. 59 "sea ice coupled models." Does coupling refer to GCMs and ESMs, or just coupling to ocean models?

**A5**: Here we focus on the coupling of ice model and ocean model.

l. 61 This is the first use of the term "monotonic" in the main text. Here I suggest defining monotonicity in the context of sea ice transport. Typically, this term refers to schemes that don't introduce spurious new maxes or mins in tracers such as ice thickness or enthalpy. For the case of ice concentration, it would refer to schemes that allow new maxes or mins only when the velocity field is convergent or divergent.

**A6**: Thanks for the advice, and you conjecture is consistent with our motivation. Firstly, we use the FEM-FCT scheme of single class ice model in SCHISM, but Icepack always aborts, and we find there is unphysical tracers. So we want to make a tracer monotonic scheme, and we have defined the monotonicity in l. 82 of the track-changes file as:
For the sea ice model, the monotonicity means the new tracers will not exceed the local maximum or minimum around it (Lipscomb and Hunke, 2004). And for ice concentration, it can exceed 1 and leads to ridge which has been described in Icepack.

l. 63 "Lipscomb et al. (2004)." Should be "Lipscomb and Hunke (2004)"

**A7**: Thanks for the reminder, and we have corrected them.

This part has been revised to 'Lipscomb and Hunke (2004) implemented' in l. 90 of the track-changes file.

l. 67 I don't understand the claim that incremental remapping (IR) is inefficient for unstructured grids. The geometric part of the IR computation scales linearly with the number of grid cells, and the tracer-reconstruction part scales super linearly with the number of tracers. For CICE and MPAS-Sea ice users, the cost of transport is typically not greater than the cost of EVP dynamics and Icepack column physics. If the SCHISM developers opted for TVD in favor of IR, it likely wasn't for reasons of computational efficiency alone. Perhaps they wanted a scheme that was easier to code?

**A8**: We apologize for the unclear statement. In Lipscomb and Hunke (2004) and Turner et al. (2022), they all say that the maximum time step may have to be reduced to ensure that trajectories do not cross. For fine mesh in some complex area, like rivers and fjords, there may be some highly distorted unstructured grid, which leads to frequent cross trajectories,

so excessively smaller time step    is needed and it will reduce the efficiency. We have detailly describe this issue instead of using the word 'inefficient' in the revised paper.

This part has been revised to 'The incremental remapping scheme is a second-order accurate scheme, and has great performance in structured grid models, but requires excessively smaller time step to avoid cross trajectories for highly distorted UGs.' in l. 94 of the track-changes file.

p. 76    "It is unclear…". I don't know why it would be unclear whether or not a scheme is monotonic.

A9: In Gao et al. 2013, they call the transport as the second-order upwind scheme, but they do not explain in datil, and they do not demonstrate the monotonicity. In the revise version, we have tried to reproduce the result of their method, but it is more diffusion than the TVD scheme (in the general answer), so we decide not to use second-order upwind scheme to demonstrate its monotonicity.

l. 80 For many transport schemes (IR is an exception), the cost increases linearly with the number of variables. Likewise, there is always some cost (usually justified) to imposing strict monotonicity. I'm not sure why these were reasons to rule out FEM-FCT.

A10: For FEM-FCT, the computational cost is twice as much as normal finite element Taylor–Galerkin scheme as they add an anti-diffusive term, and in general, it also needs sub-cycle to satisfy the Courant condition. So when it is used to multi-class ice model, the increase in the computational cost can be substantial. And it is more important that FEM-FCT is not monotonic as we shown in the general answer.

l. 81 Here the authors describe the simplicity of the previous version of SCHISM: upwind transport, 0-layer thermodynamics, etc. They could expand on this discussion to say why there was a need for Icepack and other upgrades.

A11: Thanks for the comment. We have introduced the motivation in the revised version and shown in the general answer. In the previous version of SCHISM, the melting is faster than observation in the Great Lake case. And the FEM-FCT scheme needs to be decayed to low-order solution in their study for single-class ice module somewhere to maintain the stability.

l. 89 "The performance of the multi-class sea ice formulation has not been tested before." This suggests the value of a more complete validation as suggested above.

Answer together in A11.

l. 92    This is where TVD is introduced as a scheme with the desired properties. Can the authors define the method and say when and by whom it was introduced? Does it have a prior history in sea ice modeling?

A12: Actually, TVD is the transport scheme of ocean tracers in SCHISM and SELFE, and Zhang

et al. (2016) has updated it to TVD$^2$, which is implicit in the vertical dimension to save cost and be more accurate. In our ice model, we do not have vertical layers for ice transport, so the normal TVD scheme is appropriate.

In the revised version, we introduce as 'SCHISM is a derivative product built from the original Semi-implicit Eulerian–Lagrangian Finite Element (SELFE, v3.1dc; Zhang and Baptista 2008) with multiple enhancements, including the seamless cross-scale capability from creek to ocean, a mass conservative, monotone, higher-order transport solver TVD$^2$ (implicit TVD in the vertical and explicit TVD in the horizontal, Zhang et al., 2016). ' in l. 68 of the track-changes file.

l. 106 The ITD implementation in Icepack is based on Lipscomb (2001), not Bitz et al. (2001).

**A13**: Thanks for the reminder, we have added the Lipscomb (2001). And I also have checked with Icepack, and in source code, icepack_itd.f90, it recommends 2 paper,    Bitz, C.M., and W.H. Lipscomb, 1999: An energy-conserving thermodynamic model of sea ice, J. Geophys. Res., 104, 15,669--15,677. and Bitz, C.M., M.M. Holland, A.J. Weaver, M. Eby, 2001: Simulating the ice-thickness distribution in a climate model, J. Geophys. Res., 106, 2441--2464. We have added them all.

This part has revised to 'therefore an ice thickness distribution (ITD, Lipscomb, 2001, Bitz et al., 2001, Bitz and Lipscomb, 1999)' in l.147 of the track-changes file.

l. 120 What is meant by "hydrodynamic core"? Is this the ocean model, or is it something more general than an ocean model?

**A14**: Yes, and there are many modules in SCHISM. SCHISM has capability to simulate situation across creek-lake-river-estuary-shelf-ocean scales, so it may not only an ocean model. The hydrodynamic core includes hydrostatic solver, boundary condition and so on.

l. 137 I'm not sure it's accurate to say that transport is "the main challenge." Thermodynamics and ridging are challenging too.

**A15**: We sincerely agree that thermodynamics and ridging are challenging too. But they have been considered in Icepack, and we have coupled them. What we should do is the dynamic parts for the coupled model.

p. 141 "a strictly monotone scheme is still desirable". See the l. 61 comment; it would be better to define and discuss monotonicity earlier.

**A16**: Thanks, and we have reorganized the Introduction.

This part has been revised to 'In the sea ice model, monotonicity ensures that the values of new tracers do not exceed the local extrema, specifically the maximum or minimum values in their vicinity under pure advection (Lipscomb and Hunke, 2004). For ice concentration, it can exceed 1 and results in ridge which has been described in Icepack.' in l. 82 of the track-changes file.

l. 143 It would be better to introduce the SCHISM model and grid earlier. Please say what is meant by an Arakawa CD-grid. Does the first use of "SCHISM" on l. 144 refer to the SCHISM lake/ocean component?

**A17**: Arakawa CD-grid is the grid that traces locate at the node while the velocity locates at the side. The "SCHISM" on l. 144 refer to the SCHISM lake/ocean component and we have revised it as 'The sea ice module inside SCHISM employs an Arakawa-A grid, with both the sea ice velocity and tracers located at the node (blue circles in Fig.1).'. And the discussion of grid definition is moved from Section 2.2 to Section 2.1, as 'The ice module uses the Arakawa-A grid, and all tracers and velocities are defined at nodes, while the hydrodynamic module uses the Arakawa-CD grid. The decision to employ an analogue of the Arakawa-A grid in the rheology part, adapted from FESIM, was primarily based on its computational efficiency and success in sea ice simulation (Danilov et al., 2015). ' in l. 165 of the track-changes file.

l. 146 "centroids". Meaning centroids of triangles, as opposed to centroids of hexagons?

**A18**: It is the centroid of triangle, as the red dot in Fig.1 of manuscript.

l. 163 It might be helpful to give the reader some examples of Eqs. (7) and (8) in action, showing how they work to preserve monotonicity. For example, one could consider the three cases of (phi_C – phi_U*) = (phi_D – phi_C), 0, and -(phi_D – phi_C), which would imply psi_i = 1, 0, and 0, respectively.

**A19**: Thanks for the suggestion, we explain it with more details. And this part has been revised to 'If $r_i < 0$, it means $\phi_C$ is a local extreme, $\phi_i$ in Eq.6 will revert to upwind. If $r_i > 0$, there is no local extreme, so $\phi_i$ is a weighted average of $\phi_C$ and $\phi_D$.' in l. 218 of the track-changes file.

l. 169 "gradient of the central node grad(phi_C)". Does this mean the quantity grad(phi), evaluated at node C? How is the gradient evaluated? E.g., with a line integral around the adjacent nodes?

**A20**: In SCHISM, the gradient of element is an existing variable, so the gradient at the node C is the weighted average value by element area around the node C.

l. 181 I'm not sure phi_U* is meant here, since it's not an edge tracer value. Should this be phi_i, as computed in Eq. 6?

**A21**: Sorry for the mistake. It is the $\phi_i$ in Eq.6 in manuscript indeed.

This part has been revised to 'Using the approximation of edge tracer values $\phi_i$, we can calculate the sea ice area fluxes across every edge of the control volume' in l.197 of the track-changes file.

l. 181    The term "sea ice fluxes" is ambiguous. Does this mean fluxes of ice area?

**A22**: It is the ice area flux, and we have modified it.

This part has been revised to 'we can calculate the sea ice area fluxes across every edge of the control volume' in l. 236 of the track-changes file.

l. 182 Is van Leer limiting applied to h and q? I think it must be, if h and q are to be advected monotonically.

A23: The van Leer limiting is not applied to them. For h and q, we use upwind scheme based on ice area flux. And we will test the case with the limiter applied to tracers in further work.

l. 205 "Since the thermodynamic part…". Icepack is a significant step forward for SCHISM, so it would be interesting to know if it improves results compared to earlier model versions, for either the Great Lakes or the Arctic.

A24: Thanks for the suggestion, and we to compare the results of these two versions. The results are shown in the general answer.

l. 211. A time step of 1 s seems unnecessarily short given the size of the triangles (200 m on a side) and speed of the flow (1 m/s).

A25: For general CFL condition of transport, the time step is too small, but in SCHISM, we always add tidal component, so the recommend CFL condition in hydro module is

$$CFL = \frac{(|u| + \sqrt{gh})\Delta t}{\Delta x}$$

Another reason is that when the horizontal scale is closed to vertical scale and dt is also large, there is spurious 'upwelling' in SCHISM as it described in SCHISM manual, so we need to reduce the dt.

l. 213 It is predictable that TVD will outperform upwind and centered difference schemes in exactly the ways described. There is no need to include conservation as a metric, since TVD (like upwind) is conservative by construction, whereas centered is not (given that over- and undershoots are clipped). Thus, the following three sections (on accuracy, conservation, and monotonicity) are longer than necessary and not very illuminating.

A more relevant analysis would be to compare TVD to incremental remapping (if the authors were able to set up similar test problems in CICE or MPAS-Seaice) or another second-order monotone scheme. In the case of IR, it could be interesting to show that TVD gives similar results at lower cost.

A26: In this part, we want to show that the TVD has the property of accuracy, conservation, and monotonicity. And we compare TVD with FEM-FCT and second-order upwind of FVCOM in the revised version and the results are shown in the general answer.

l. 289 The high-resolution Great Lakes simulation is a good problem for comparing TVD and upwind. The results shown in Fig. 6 are quite convincing. For this reason, I think the authors

could leave out the simple problems in Section 3.1 and let Section 3.2.1 make the case for TVD over upwind.

It is interesting that the TVD method reduces the overall model cost (compared to upwind) by limiting diffusion of ice area. How much time is spent in the transport solver alone for each of the two transport schemes, and how does this compare to the total model time?

**A27**: We agree with the suggestion that the high-resolution Great Lakes simulation is a good problem to compare TVD and upwind.    And in the revised version, we decide to compare TVD to two other second-order accurate schemes still in section 3.1, and keep the section 3.2 to demonstrate the efficiency of TVD with the high-resolution Great Lakes simulation. We also have counted the total model time of ice module. This part has been revised to 'the total simulation times for the two schemes are comparable, 678 minutes for the TVD scheme and 675 minutes for the upwind scheme. In total, the upwind scheme consumes 52.39 core hours while TVD spends 54.56 core hours. Compared to the total time of the ice module, TVD accounts for 21.71% and upwind accounts for 21.01%, while the dynamic part is the most computationally intensive, accounting for more than 70%. Overall, we found that the computational cost of the TVD scheme is comparable to that of the upwind scheme in this realistic benchmark application.' in l. 390 of the track-changes file.

l. 304 I suggest "compare" instead of "qualitatively compare", since the comparison is not merely qualitative.

**A28**: Thanks for the suggestion. We have compared the results with observation data.

This part has been revised to 'The spatial distribution of the ice concentration from the two models is compared with the observation from the U.S. National Ice Center (USNIC) on day 90 (Fig.7), when the ice cover was largest.' in l. 421 of the track-changes file.

l. 315 Since this section focuses on general model validation (rather than a validation of TVD), I suggest expanding it and making it an entire section rather than a subsection. Also, it would be useful to see how the new model version compares with the older, simpler version.

**A29**: This section talks about a realistic case to evaluate the whole model performance. The update of Icepack is significant as multi-class ice model is more advanced than the single class ice model, and the version has better performance than the older version. We also have compared the transport scheme of single class, FEM-FCT, in an idealized case to demonstrate what the advance of TVD is. All these results have been shown in the general answer.

l. 318    Is the sea ice time step just 100 s? This seems unnecessarily short if the minimum grid cell size is 6 km, assuming a max speed of ~1 m/s.

Answer together in A25.

l. 328    I am not sure what is meant by "the generic length-scale equation as k-kl."

**A30**: It is a generic length-scale model with a k-kl configuration, and the different configuration has different parameter, like different generic length-scale variable.

l. 330 How is TVD2 related to the TVD scheme implemented for the sea ice model? Answer together in A12.

l. 341 "which may be influenced by the initial conditions as we did not get all tracers, such as sea ice salinity and enthalpy, from HYCOM." I can think of many reasons why the first peak might not line up with the observed value. I'm not sure why initial tracer values are singled out as an explanation.

**A31**: We also agree that there are many reasons for the error, but the first peak is higher than others. So we think it may be induced by some systematical error, which is the incomplete initial conditions.

l. 350 Why was FESOM2, as opposed to some other model, chosen as a standard for comparison? How similar was the FESOM2 configuration?

**A32**: The single class ice model of SCHISM is borrowed from FESOM, and when we develop the multi-class ice module of SCHISM, we refer the framework of FESOM2(Zampieri et al., 2021)). Another reason is both SCHISM and FESOM2 are the ocean model coupling with Icepack on unstructured grid.

l. 354 "the simulated sea ice extent often increases faster in autumn than observation." This isn't obvious from Fig. 7a. I just see one year (1994) when the modeled September min is significantly greater than observed.

**A33**:   There is only one blue dot(model) which is greater than the orange dot(observation) in September, and all others values are close. But in October and November, blue dots are always greater than oranges. So, we think the increasing is faster in autumn.

l. 361 Typically when comparing two models, one would force them over the same integration period. If FESOM runs are available from 1994–1999, I would suggest using those. If not, then it might be better to leave out the comparison.

**A34**: Thanks for the suggestion. Two models have different integration period, but what we want to compare is the bias of the month average from observation. Although the bias may be induced by forcing, it also can demonstrate the accuracy of the coupled model in some degree.

l. 368 It's helpful to see these spatial patterns of sea ice concentration biases. Would it be possible also to show plots of sea ice thickness compared to observations?

**A35**: Thanks for the suggestion, and we have found some thickness data based on submarine from NSIDC and the results are shown in the general answer.

l. 371    Do the authors know why the ice edge is too far advanced on the Atlantic side, and not far enough on the Pacific side? Is this likely an ocean model bias?

A36: Thanks for the question, and this may be some bias of the boundary condition or due to the resolution of mesh, and the ocean current seems more complex on the Atlantic side, but we have not explored the reason for it yet.

l. 381 The melt pond hypothesis is interesting. Is it possible to test this idea by, for instance, turning off melt ponds or using different precipitation forcing?

A37: Yes, and we agree that it is a valuable topic to be explored, but it is beyond the scope of this paper and we may test the idea in the further work.

l. 393 The discussion section is short and includes some material (e.g., grid choice) that would fit better earlier in the paper. It doesn't shed new light on the Section 3 results. I would suggest leaving it out.

A38: We reorganized this part and adjust the content to Section 1 and 2.

l. 405 I doubt that "remarkable" is the right word here. Again, I don't think the centered difference scheme adds value to the Section 3 analysis.

Answer with A26.

l. 410 This is an odd place to introduce the Casulli et al. scheme. Maybe do this earlier, in Section 2.2 or 3.1.

Answer with A38.

l. 417 The conclusion is short and cursory. It would be better to include a discussion of how the addition of Icepack and the TVD scheme have improved SCHISM compared to the previous model version.

Answer with A29 and A25 for the comparation to the previous model version.

l. 460 The reference list is incomplete and contains some errors. For instance, there is no Gurvan et al. (2022) or Campin et al. (2023).

A39: Thanks for the advice. We have revised.

These parts have been Revised to '(NEMO, and NEMO can also couple with CICE or The Louvain-La-Neuve sea ice model, LIM3, Madec et al., 2022)' in l. 44 and '(MITgcm; Adcroft et al., 2023).' in l. 46 of the track-changes file.

**Minor corrections**

l. 23 "the satellite" -> "satellites"
l. 25 "dramatically" -> "dramatic", "Sea ice" -> "sea ice"

l. 29 "the sea ice models" -> "sea ice models"

l. 87 "The Great Lake" -> "the Great Lake"

l. 109 (and elsewhere): "traces" -> "tracers"

l. 132 (and elsewhere): "Where" -> "where"

l. 139 (and elsewhere): Check punctuation with equations. Here, the comma should be a period.

l. 162 "Van-leer" -> "van Leer". Also l. 197.

l. 183 "as does CICE" -> "as in CICE"

l. 324    "manning" -> "Manning"

l. 325 "The" -> "the"

l. 335 "Sea Ice Concentrations" -> "sea ice concentration"

This is not a complete list. There are many minor typographical and grammatical errors that should be cleaned up in the next version.

**A**: Thanks for the advices. We have revised them.

Reference:

Gao, G., Chen, C., Qi, J., and Beardsley, R. C.: An unstructured-grid, finite-volume sea ice model: Development, validation, and application, Journal of Geophysical Research, 116, https://doi.org/10.1029/2010JC006688, 2011.

Great Lakes Surface Environmental Analysis (GLSEA): https://coastwatch.glerl.noaa.gov/satellite-data-products/great-lakes-surface-environmental-analysis-glsea/, , last access: 2 April 2024.

Lipscomb, W. H. and Hunke, E. C.: Modeling sea ice transport using incremental remapping, Mon. Weather Rev., 132, 1341–1354, https://doi.org/10.1175/1520-0493(2004)132<1341:MSITUI>2.0.CO;2,    2004.

Löhner, R., Morgan, K., Peraire, J., & Vahdati, M. Finite element flux‐corrected transport (FEM–FCT) for the euler and Navier–Stokes equations. International Journal for Numerical Methods in Fluids, 7(10), 1093-1109. https://doi.org/10.1002/fld.1650071007, 1987.

National Snow and Ice Data Center (comp.). Submarine Upward Looking Sonar Ice Draft Profile Data and Statistics, Version 1 [Data Set]. Boulder, Colorado USA. National Snow and Ice Data Center. https://doi.org/10.7265/N54Q7RWK. Date Accessed 03-23-2024. 1998.

The U.S. National Ice Center (USNIC) : https://usicecenter.gov/    last access: 2 April 2024.

Turner, A. K., Lipscomb, W. H., Hunke, E. C., Jacobsen, D. W., Jeffery, N., Engwirda, D., Ringler, T. D., and Wolfe, J. D.: MPAS-Seaice (v1.0.0): sea-ice dynamics on unstructured Voronoi meshes, Geoscientific Model Development, 15, 3721-3751, https://doi.org/10.5194/gmd-15-3721-2022, 2022.

Zampieri, L., Kauker, F., Fröhle, J., Sumata, H., Hunke, E. C., and Goessling, H. F.: Impact of Sea‐Ice Model Complexity on the Performance of an Unstructured‐Mesh Sea‐Ice/Ocean Model under Different Atmospheric Forcings, Journal of Advances in Modeling Earth Systems, 13, https://doi.org/10.1029/2020MS002438, 2021.

Zhang, Y. and Baptista, A.M. SELFE: A semi-implicit Eulerian-Lagrangian finite-element model

for cross-scale ocean circulation", Ocean Modelling, 21(3-4), 71-96. https://doi.org/10.1016/j.ocemod.2007.11.005, 2008.

Zhang, Y. J., Ye, F., Stanev, E. V., and Grashorn, S.: Seamless cross-scale modeling with SCHISM, Ocean Modelling, 102, 64-81, https://doi.org/10.1016/j.ocemod.2016.05.002, 2016.

Zhang, Y. J., Wu, C., Anderson, J., Danilov, S., Wang, Q., Liu, Y., and Wang, Q.: Lake ice simulation using a 3D unstructured grid model, Ocean Dynamics, 73, 219-230, https://doi.org/10.1007/s10236-023-01549-9, 2023.

Reply to Topic editor

**Editor**: Thank you to both reviewers and the authors for the reviews + response. Both reviewers suggest major revisions but an interesting modeling development, and I will invite the authors to resubmit their manuscript following the changes they have outlined for a second review. For consistency as well - could the authors please indicate in their abstract the version numbers of icepack/Schism they are updating.

**Response:** Hello editor, thanks for your review and help! In response to your suggestion, we have revised the abstract and added the version number as 'In this study, we couple the Semi-implicit Cross-scale Hydro-science Integrated System Model (SCHISM, v5.11) with Icepack (v1.3.4), the column physics package of the sea ice model CICE;' in line 16 of the track-changes file.

---

## Author Response (AR5)

**From topic editor,**

Thanks for your efforts here, and I am happy to accept pursuant to some modification of the response letter to see where changes were made.

Response:
We are truly grateful for your assistance and sincerely apologize for any inconvenience caused by the lack of details in the former response file. To provide further clarity regarding the modifications in the tracked changes file, we have revised the response file and included all the line numbers and changes.

**Report #1**

The revised manuscript was improved in many directions. However, there are still some points that have to be addressed.

A1: We greatly appreciate your comments!! It has greatly contributed to the improvement of our paper.

1. The authors state that the new TVD advection scheme is the one already used in SCHISM (lines 100-105). In this respect the title is misleading -- what is the development? Furthermore, the discussion of monotonicity requires some adjustments. First of all the notion of monotone scheme can only be introduced for a non-divergent velocity field. This should be clearly stated, and the text should be edited in several places where the authors introduce contradicting statements. Second, it should be clearly explained that although in sea-ice case the ice velocity is generally divergent, one prefers to use schemes that are monotone is the limit of vanishing divergence. The reasons are two-fold. One needs to maintain positivity, and one needs to suppress dispersive errors. There is no clear explanation in the manuscript at present.

A2: Although the TVD scheme in the ice module is based on the original TVD scheme of the hydro model, there are fundamental differences between the two. In the hydro model, the TVD advection scheme is implemented for tracers such as water temperature and salinity in 3-D model. It is the first time this scheme is used to transport sea ice-related tracers, and we have made efforts to convert the scheme to an explicit format for a 2-D model. Another significant difference is that in the hydro model, the TVD scheme is based on an Arakawa-CD grid, while in the ice module, it is based on an Arakawa-A grid. Significant efforts were made to develop, debug, and validate the sea ice TVD model, resulting in a mass-conservative, monotone, higher-order transport solver. Based on these substantial improvements to the TVD scheme for sea ice variables, we consider this work to be a significant development.

We totally agree that a monotone scheme for ice concentration can only be introduced for a non-divergent velocity field and will be non-monotonic when the ice velocity is divergent. Ice concentration can exceed 1 when divergence occurs, resulting in ridging, which is described in Icepack and is allowed in our model. The monotonicity we aim to guarantee is for other tracers of ice, such as ice enthalpy and ice salinity. Even in a divergent ice velocity field, these variables should remain monotonic and not exceed local extrema. However, these tracers can become non-monotonic due to numerical errors induced by improper advection schemes. As demonstrated in Figure 4, the FEM-FCT scheme causes the thickness to overshoot the initial value and oscillate at the trailing edge in a uniform velocity field, destabilizing the realized case. We also wanted to show patterns of other tracers, such as salinity, but due to FEM-FCT's instability with Icepack, we chose ice thickness as the representative tracer for both the single-class and multi-class ice models. For the ice velocity, we do not impose any limits to eliminate divergence and follow the mEVP of FESOM faithfully. Therefore, when the ice velocity is divergent, the ice concentration will be non-monotonic, but other tracers, like ice salinity, should still be monotonic, which is our goal and what we are working towards. We sincerely appreciate the reviewer's suggestion and have revised several parts related to monotonicity, which

are listed in the minor points.

2. I was suggesting in the first review that the authors will present more details in the manuscript explaining how monotonicity is achieved (develop (3) using (4) - (8)). It will be then clearly illustrated that the scheme is monotone for non-divergent velocities, but not monotone for diverging velocities. This is common property.

A3: We apologize for any confusion caused by our initial explanation. In Eq.3 – Eq.9, all $\phi_i$ in these equations represent ice concentration. Therefore, for divergent velocities, the concentration can be non-monotonic, as we mentioned previously in response A2. Other tracers, like ice salinity, derived by Eq.10-14, are guaranteed to be monotonic. Because we treat them as an essentially weighted average method with non-negative weights. We have revised the manuscript to include additional details that illustrate the $\phi_i$ in line 175 of the track-changes file:

Most of these variables can be obtained easily in the model, so we only focus on finding a method to approximate the edge value, $\phi_i$ (this symbol always represents ice concentration hereafter).

Some other points are mentioned below.

Line 15 'more advanced' -- does not tell anything, please be specific.

A4: 'more advanced' here means the scheme should meet the requirements in L.21 which includes conservation, accuracy, efficiency, and strict monotonicity for tracers. In order to avoid the duplication of discussion, we prefer to stating that we need an evolved transport scheme here. So we have revised as in line 14 of the track-changes file:

As the demand for increased resolution and complexity in unstructured sea ice models is growing, higher demands are also placed on sea ice transport scheme.

20 'better performance' -- does not tell anything. It is not clear from the abstract what is the problem, and mentioning 'strict' monotonicity without explaining in which context this notion is used only creates a problem as everybody knows about ridging, i.e. violation of strict monotonicity.

A5: 'better performance' here means TVD scheme meets the requirements for conservation, accuracy, efficiency (even with very high resolution), and strict monotonicity for tracers (like the ice thickness and enthalpy, but not include concentration). In single class ice model of SCHISM, the FEM-FCT is satisfied with these requirements with some minor modifications in Zhang et al. (2023). When we developed the multi-class ice model with SCHISM, the FEM-FCT is always unstable with Icepack. As we stated in Conclusion,' The simulation results reveal that the TVD scheme is conservative, accurate, strictly monotonic, and efficient in reproducing the horizontal transport of ice, and has better accuracy than the second-order upwind scheme at similar computational cost. Particularly, it provides strict monotonicity, which is crucial for stability, thus addresses the difficulties encountered in the single-class ice model utilizing the FEM-FCT.' So we called that TVD has better performance than FEM-FCT and second-order upwind. We have explained the monotonicity more clearly in A2 and this part has been revised with more details as in line

19 of the track-changes file:
Compared with the second-order upwind scheme and the Finite Element Flux Corrected Transport (FEM-FCT) scheme, the TVD transport scheme is overall superior when evaluated based on conservation, accuracy, efficiency (even with very high resolution), and strict monotonicity. Although it is slightly weaker than FEM-FCT in terms of accuracy alone, the TVD scheme still outperforms the other two schemes in comprehensive performance.

Line 75 This is an example when monotonicity is mentioned, but on the next line it is said that it is not working, which only irritate your reader, see my comment 1.
A6: We are sorry for the unclear statement, the explanation has been listed in A2, and we have revised as in line 79 of the track-changes file:
In the sea ice model, monotonicity ensures that the values of new tracers, such as ice thickness and enthalpy (but not ice concentration), do not exceed the local extrema, specifically the maximum or minimum values in their vicinity under pure advection (Lipscomb and Hunke, 2004), even when ice concentration exceeds 1 and results in ridge which has been described in Icepack.

line 82-84 are still the author's interpretation, which is inappropriate. (i) The CFL criterion will always limit time steps on highly distorted meshes for explicit schemes. (ii) MPAS-Seaice is formulated on hexagonal meshes. Triangular meshes dual to their hexagonal meshes are of high quality because they are orthogonal (circumcenters of triangles are inside triangles). MPAS-Seaice can operate on any resolution.
A7: We agree that MPAS-Seaice can operate on any resolution and the time step should be limited on highly distorted meshes for explicit schemes. In MPAS, Turner et al. (2022) stated 'the time step is limited by the requirement that trajectories projected backward from vertices are confined to the cells sharing the vertex', and 'For highly divergent velocity fields, the maximum time step may have to be reduced by a factor of 2 to ensure that trajectories do not cross'. While in SCHISM, the model is very forgiving in mesh quality, one of the reasons is the TVD transport scheme. In light of these considerations, and to facilitate the use of this model on complex unstructured grid meshes, we aim to retain the operational efficiency of SCHISM, even for highly distorted unstructured grids in the context of sea ice modeling. Taking your comments into account, we will make the following changes in line 86 of the track-changes file:
The incremental remapping scheme is a second-order accurate scheme, and has great performance in structured grid models and MPAS-Seaice, but requires excessively smaller time step to avoid cross trajectories when the velocity field is divergent (Lipscomb and Hunke, 2004) or for highly distorted UGs (Turner et al., 2022).

95 What do the authors mean under the high cost? By construction the scheme by Loehner et al. is monotone. The discussion further is again strange, as it is not clear what is meant and why there are problems.
A8: We are sorry for the unclear statement. Initially, we attempted to address the non-monotonicity issues in the FEM-FCT method; however, these efforts were unsuccessful. In Loehner et al. (1987), they said the low-order scheme in any FCT-method should be

monotonicity, but the obvious candidate, Godunov's method, is more expensive, so they chose the Taylor-Galerkin scheme which is least expensive and added mass-diffusion to guarantee the monotonicity. However, they encountered unphysical negative pressures in their numerical examples using this approach. So they added some additional limiter to keep positive pressure artificially. Based on our investigations, we believe that the non-monotonicity pattern may be attributable to the low-order Taylor-Galerkin scheme. Implementing the more expensive Godunov's method would likely ensure monotonicity, despite its higher computational cost.

105 'the new TVD scheme' -- Is it new? See your line 103
A9: Answer together in A2.

lines 159-161 monotonicity again
A10: Answer together in A2 and has been revised as in line 165 of the track-changes file: Note that the ice velocity field is divergent or convergent, which can produce new local maxima/minima for $a_n$. However, a strictly monotone scheme is still desirable in order to separate the numerical dispersion from the physical convergence, especially for tracers like ice enthalpy and salinity.

169 approximate
A11: Thanks for pointing out, we have corrected it as in line 175 of the track-changes file: Most of these variables can be obtained easily in the model, so we only focus on finding a method to approximate the edge value, $\phi_i$

180-181 explain the weighted average, your reader does not see this, because (5) contains difference of these two values.
A12: In Eq. (5)

$$\phi_i = \phi_C + \frac{\psi_i}{2}(\phi_D - \phi_C),$$

it is same as

$$\phi_i = \frac{2-\psi_i}{2}\phi_C + \frac{\psi_i}{2}\phi_D,$$

while $\psi_i \in [0,2)$, so we called it is a weighted average. We have revised as in line 188 of the track-changes file:
If $r_i < 0$, it means $\phi_C$ is a local extreme, $\phi_i$ in Eq.6 will revert to upwind. If $r_i > 0$, there is no local extreme, and $\psi_i \in [0,2)$, so $\phi_i$ is a weighted average of $\phi_C$ and $\phi_D$ in Eq.5.

Formula (9) is only valid for concentration, but has to be changed for other quantities.
A13: Answer together in A3. And Formula (9) is only valid for concentration indeed. We have removed 'tracer' around Eq.9 for clarification in line 170 and 206 of the track-changes file:
The control volume is defined as the polygon enclosed by the lines composed of centroids and edge centers (red circles in Fig.1).

Using the approximation of edge values $\phi_i$, we can calculate the sea ice area fluxes across every edge of the control volume, and thus the new concentration from Eq. (3).

218 'non-negative weights' -- help your reader to see this.
A14: We are sorry for the unclear statement. According to the note in the previous reply, we have revised manuscript as in line 227 of the track-changes file:
Furthermore, the monotonicity of tracers is guaranteed because the method in Eq. (11) and Eq. (13) is essentially a weighted average method with non-negative weights. And in general, the exchange caused by advection are relatively small in amount, $a_n^t \gg \frac{\Delta t \sum_{i \in S} Q_i \phi_i}{\Omega_S}$, so the non-negativity of $a_n^{t+1}$ is guaranteed. When we consider a divergent flow, the $h_i$ of Eq. (11) is just equal to $h_n$, the centre node value and here is

$$v_n^{t+1} = a_n^t h_n + \frac{\Delta t \sum_{i \in S} Q_i \phi_i}{\Omega_S} h_n,$$

which is always non-negative.

Fig. 2: What is called second-order upwind seems to me to be even worse than the first-order upwind scheme, and I therefore find the result strange. The scheme described in Gao et al. is not monotone, and its dissipative truncation error is fourth-order. It has third-order dispersive errors, and should show oscillations, as these errors are dominant. The associated biharmonic dissipation is not as strong as in Fig. 2. Please check carefully, something is wrong.
A15: We also found this result surprising. To reproduce the second-order upwind scheme, we thoroughly reviewed both the article and the accompanying code. And faithfully we follow the code, including the implementation of the upwind control volume and the calculation of the gradient at the centroid. The discrepancy between our results and those reported by Gao et al. (2011) might be due to differences of gird. Although tracers are positioned at vertices (nodes) similar to our model, the velocity is calculated at the centroids, differing from our scheme.

FEM-FCT is noticeably more accurate than the proposed scheme (use, e.g., L2 norm to see this).
A16: While we acknowledge that FEM-FCT is slightly more accurate than the TVD scheme, the difference in accuracy is minimal. However, the TVD scheme ensures strict monotonicity in a non-diffusive field, which is crucial for maintaining the model's stability.

263 This statement contradicts the construction of this scheme in Loehner et al. It should be related to some issues of the implementation, but should not be a property of the scheme.
A17: Answer together in A8.

Figure 4. Please explain what is shown in this figure. I still cannot understand why 1.5 m

pulse is stretched to occupy larger spatial extent.

A18: The phenomenon likely results from numerical diffusion. In Fig.2 and Fig.3 of manuscript, we only showed the pattern of its concentration greater than 15%. And in Fig.4 of manuscript, the threshold was set significantly lower, at 0.1%. If we set the threshold of ice concentration to 1%, the concentration and volume per unit area would still exhibit a larger spatial extent, as shown in the figure below. This pattern is consistent across all the schemes we analyzed, which further supports the hypothesis that numerical diffusion is responsible for the observed stretching.

[Figure]

Figure.1 the snapshot of ice concentration, volume per unit and thickness from FEM-FCT

To clarify the difference between fig.2 and fig.4, we have added more explanation in line 294 of the track-changes file:
Considering that non-monotonicity typically occurs in areas of low ice concentration, we choose 0.1% as the threshold rather than the previous 15%

286 'We demonstrated' -- No demonstration of the second-order convergence is proposed in the manuscript. As I've written above, the second-order upwind scheme does not look as the second-order (no dispersive errors), so I suspect an issue in its implementation.
A19: Answer together in A15.

291 Again 'higher cost' without explanation what is meant.
A20: Answer together in A8.

445 The statement here will sound strange unless the author explain the source of difficulties -- the Loehner et all scheme is monotone provided the time step is limited. Please show which part of the algorithm is leading to problems.
A21: Answer together in A8.

In the end, I see the advantage of the new scheme in its lower cost compared to FEM-FCT, which might be important in practice given the use of ICEPACK and the need to transport multiple tracers. The other argument can be its (presumably) larger admissible time step.

The attempt of the authors to motivate the need for their 'new' (not really) scheme from the monotonicity consideration is unfortunate in my opinion.

A22: Thank you for your comments. In summary, we use the TVD scheme because it enhances the model's stability due to its monotonicity and offers relatively good accuracy. We hope this explanation clarifies our choice and thereby makes the manuscript easier to understand.

Reference:

Gao, G., Chen, C., Qi, J., and Beardsley, R. C.: An unstructured-grid, finite-volume sea ice model: Development, validation, and application, Journal of Geophysical Research, 116, https://doi.org/10.1029/2010JC006688, 2011.

Löhner, R., Morgan, K., Peraire, J., & Vahdati, M. Finite element flux‑corrected transport (FEM–FCT) for the euler and Navier–Stokes equations. International Journal for Numerical Methods in Fluids, 7(10), 1093-1109. https://doi.org/10.1002/fld.1650071007, 1987.

Lipscomb, W. H. and Hunke, E. C.: Modeling sea ice transport using incremental remapping, Mon. Weather Rev., 132, 1341–1354, https://doi.org/10.1175/1520-0493(2004)132<1341:MSITUI>2.0.CO;2, 2004.

Turner, A. K., Lipscomb, W. H., Hunke, E. C., Jacobsen, D. W., Jeffery, N., Engwirda, D., Ringler, T. D., and Wolfe, J. D.: MPAS-Seaice (v1.0.0): sea-ice dynamics on unstructured Voronoi meshes, Geoscientific Model Development, 15, 3721-3751, https://doi.org/10.5194/gmd-15-3721-2022, 2022.

Zhang, Y. J., Wu, C., Anderson, J., Danilov, S., Wang, Q., Liu, Y., and Wang, Q.: Lake ice simulation using a 3D unstructured grid model, Ocean Dynamics, 73, 219-230, https://doi.org/10.1007/s10236-023-01549-9, 2023.

**Report #2**

The authors have responded to all the major critiques and most of the minor suggestions from the first round of reviews. The new version clearly explains why TVD is a suitable advection scheme for the coupled SCHISM–Icepack model, given the non-uniform unstructured mesh. I like the new analysis in Section 3.1, comparing TVD to the FEM-FCT and second-order upwind schemes instead of the centered and first-order upwind schemes. Section 3.2, which presents the Lake Superior and Arctic Ocean test cases, is more complete and easier to follow. I think the paper is nearly ready for publication.

A1: Thank you for your review comments. With your help, this article has significantly improved.

I suggest the following minor edits and corrections:

L. 24: Here and elsewhere, please remove "the" before "Lake Superior". This is one of several places where the paper would read better with some light editing for idiomatic English.

A2: We appreciate you pointing this out and have revised them in line 26 of the track-changes file:

The new coupled model outperforms the existing single-class ice model of SCHISM in the case of the Lake Superior.

In line 316 of the track-changes file:

SCHISM-Icepack, in conjunction with the TVD scheme for its ice transport module, is employed to reproduce the ice processes in Lake Superior and the Arctic Ocean (Fig.5).

In line 469 of the track-changes file:

The coupled SCHISM-Icepack model improves the results of the previous single-class ice model in the case of Lake Superior, and was able to reproduce the Arctic Sea ice concentration, boundary, extent, and thickness as seen from the observation.

And in line 487 of the track-changes file:

The input data of the realistic case on Lake Superior is available from Y. Joseph Zhang on reasonable request.

L. 69: What kind of model is SELFE? It isn't obvious from the acronym (Semi-implicit Eulerian–Lagrangian Finite Element).

A3: SELFE is also an ocean or hydro model, which has cross-scale capability like SCHISM, while SCHISM has multiple enhancements compared to it.

L. 96: Change "excessively smaller" to "an excessively small"

A4: Thanks for the suggestion and we have corrected it in line 86 of the track-changes file: The incremental remapping scheme is a second-order accurate scheme, and has great performance in structured grid models and MPAS-Seaice, but requires excessively small time step to avoid cross trajectories when the velocity field is divergent (Lipscomb and Hunke, 2004) or for highly distorted UGs (Turner et al., 2022).

L. 130: Change "accuracy" to "accurate"

A5: Thanks for the suggestion and we have corrected it in line 106 of the track-changes file:

The coupled model utilizes the TVD transport scheme, which has been implemented in SCHISM for ocean tracers (Zhang et al., 2016), to achieve an efficient, strictly monotone, second-order accurate scheme for ice tracers on generic unstructured grids (even with locally very high resolution).

L. 147: I think the BL99 reference is not needed here. That paper focuses on thermodynamics, not the ice thickness distribution.

A6: Thanks for the reminder, we have removed it in line 121 of the track-changes file:

At the sub-grid scale, thin and thick ice coexist, and therefore an ice thickness distribution (ITD, Lipscomb, 2001; Bitz et al., 2001) has been implemented in order to describe the unresolved spatial heterogeneity of the thickness field.

L. 167: Please say where the variables are located on the Arakawa-CD grid.

A7: Thanks for the suggestion and we have revised as in line 137 of the track-changes file:

The ice module uses the Arakawa-A grid, and all tracers and velocities are defined at nodes. The hydrodynamic module uses the Arakawa-CD grid, with velocities defined at the side centers and tracers at the prism centers.

L. 203: Change "proximate" to "approximate"

A8: Thanks for the suggestion and we have corrected it in line 175 of the track-changes file:

Most of these variables can be obtained easily in the model, so we only focus on finding a method to approximate the edge value, $\phi_i$

L. 224: Use vector notation to distinguish vectors from scalars (e.g., boldface for R_DU and R_CD)

A9: Thanks for the suggestion and we have corrected it by bolding the vector in line 195 and 196 of the track-changes file:

$$\phi_{U*} = \phi_D + \boldsymbol{R}_{DU} \cdot (\nabla\phi_C) = \phi_D - 2\boldsymbol{R}_{CD} \cdot (\nabla\phi_C), \tag{8}$$

where $\boldsymbol{R}_{DU}$ is the vector from the downwind node to the up-upwind node, and $\boldsymbol{R}_{CD}$ is the vector from the upwind to downwind nodes.

L. 233: What is meant by "Icepack will perform clipping"? Does this just mean that ridging will reduce the ice concentration to a value <= 1?

A10: Yes. In Icepack, when concentration exceeds 1 after transport, the ice will be compressed and thickened in ridge step, the concentration will recover to 1, so we called it clipping.

L. 267: Delete "part" after "thermodynamic"

A11: Thanks for the suggestion and we have corrected it in line 236 of the track-changes

file:
Since the thermodynamic and dynamic parts of this model are relatively mature and have been widely utilized in other models, in this study we focus on validating the new transport scheme.

L. 273: The author response explains why the time step is so short, but readers might still wonder about this. Please add a brief explanation in the text.
A12: Thanks for the suggestion and we have added some explanation in line 243 of the track-changes file:
The time step is 1 second, which satisfies the Courant-Friedrichs-Lewy (CFL) condition of for TVD and meets the stricter CFL condition for SCHISM (Zhang et al., 2016).

Fig. 2e: The panels in the upper part of this figure are very small and are hard to interpret. In particular, it is hard to see the banded distribution described at l. 295. Please reformat the figure in a way that better illustrates the advantages of TVD described in the text.
A13: Thanks for the suggestion and we have reformatted it by breaking axes and zooming first 3 and last 3 graph in fig.2 (has shown below and revised in the manuscript), fig.3 (has revised in the manuscript) of the track-changes file. We also add more explanation in line 252, 272, 290 of the track-changes file:
To show more details, only the first three and last three snapshots are shown.

[Figure]

[Figure]

L. 312: "while" isn't the right word here. Better wording might be "...with a peak ice volume per unit area of only 0.3 m…".

A14: Thanks for the suggestion and we have corrected it in line 274 of the track-changes file:

Among the tested schemes, the second-order upwind scheme is the most diffusive one, with the peak of ice volume per unit area of only 0.3 meters at the end.

L. 361 and Fig. 4: Please reformat the figure so that it's easier to see the oscillations at the trailing edge.

A15: Thanks for the suggestion and we have reformatted it by set y-axis to 1-2 as below and also showed in fig.4 of the track-changes file:

[Figure]

L. 390: December 1, not December 1st. Similarly at l. 448.

A16: Thanks for the suggestion and we have revised them.

In line 324 of the track-changes file:

We simulate the case for 180 days from December 1st, 2017, using 60 processors.

And in line 370 of the track-changes file:

The model starts on January 1, 1994, and covers 2000 days, about 1.6 million steps using a time step of 100 sec.

L. 411: How is the correlation coefficient computed? Is this the fraction of cells that have the same state (either ice-covered or ice-free) in both the model and the data? Please say what is meant by a Wilmot score.

A17: Both the correlation coefficient and the Wilmot score are used to evaluate the ice extent, the correlation coefficient the correlation between observed extent and simulated extent. The Wilmot score, which also used in Zhang et al.(2023) is a statistical measure used to evaluate the performance of a forecasting model and the value closer to 1 is better.

And we have revised to in line 340 of the track-changes file:

With the multi-class ice model and the TVD scheme, we are able to reproduce the similar pattern of ice extent and also some rapid melting-refreezing events, yielding a correlation coefficient of 0.93 and a Wilmot score of 0.92 (both values closer to 1 are better, Fig. 6).

And in line 344 of the track-changes file:

After the observed ice extent falls below 10,000 km², the correlation coefficient between simulated extent and observed extent with the multi-class ice model is 0.82, which is an improvement over the single-class ice model's coefficient of 0.43.

L. 426: The word "however" doesn't fit here.

A18: Thanks for the suggestion and we have removed it in line 353 of the track-changes file:

Both models exhibit lower ice concentration in the southern part of the lake; however, while in most other areas, particularly in the western region, the multi-class ice model displays lower ice concentrations.

L. 456: Delete "better"

A19: Thanks for pointing out and we have removed it in line 375 of the track-changes file:

In the vertical dimension, a highly flexible vertical gridding system (LSC$^2$, Zhang et al., 2015) is implemented with up to 60 layers in order to more accurately represent the complex topography of the Arctic Basin, and we set the bottom drag coefficient with a constant Manning coefficient of 0.0025.

L. 577: I suggest changing "performance" to "accuracy", since performance might be misinterpreted as referring to computational efficiency. Maybe reword as "and has better accuracy than the second-order upwind scheme at similar computational cost".

A20: Thanks for the suggestion and we agree that and have revised it in line 465 of the track-changes file:

The simulation results reveal that the TVD scheme is conservative, accurate, strictly monotonic, and efficient in reproducing the horizontal transport of ice, and has better accuracy than the second-order upwind scheme at similar computational cost.

Reference:

Zhang, Y. J., Ye, F., Stanev, E. V., and Grashorn, S.: Seamless cross-scale modeling with SCHISM, Ocean Modelling, 102, 64-81, https://doi.org/10.1016/j.ocemod.2016.05.002, 2016.

Zhang, Y. J., Wu, C., Anderson, J., Danilov, S., Wang, Q., Liu, Y., and Wang, Q.: Lake ice simulation using a 3D unstructured grid model, Ocean Dynamics, 73, 219-230, https://doi.org/10.1007/s10236-023-01549-9, 2023.